# Boating- and Shipping-Related Environmental Impacts and Example Management Measures: A Review

**Troy A. Byrnes** [1] 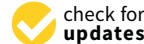 **and Ryan J. K. Dunn** [2,*]

1   5/66 Surf Parade, Broadbeach, QLD 4218, Australia; troy69byrnes@yahoo.com.au
2   Ocean Science & Technology, RPS, P.O. Box 5692, Gold Coast MC, QLD 9726, Australia
*   Correspondence: ryan.dunn@rpsgroup.com

**Abstract:** Boating and shipping operations, their associated activities and supporting infrastructure present a potential for environmental impacts. Such impacts include physical changes to bottom substrate and habitats from sources such as anchoring and mooring and vessel groundings, alterations to the physico-chemical properties of the water column and aquatic biota through the application of antifouling paints, operational and accidental discharges (ballast and bilge water, hydrocarbons, garbage and sewage), fauna collisions, and various other disturbances. Various measures exist to sustainably manage these impacts. In addition to a review of associated boating- and shipping-related environmental impacts, this paper provides an outline of the government- and industry-related measures relevant to achieving positive outcomes in an Australian context. Historically, direct regulations have been used to cover various environmental impacts associated with commercial, industrial, and recreational boating and shipping operations (e.g., MARPOL). The effectiveness of this approach is the degree to which compliance can be effectively monitored and enforced. To be effective, environmental managers require a comprehensive understanding of the full range of instruments available, and the respective roles they play in helping achieve positive environmental outcomes, including the pros and cons of the various regulatory alternatives.

**Keywords:** anti-fouling treatments; benthic disturbance; coastal environments; environmental management; hydrocarbon pollution; MARPOL; species translocation; sewage discharge; water quality; wildlife impacts

## 1. Introduction

Aquatic biomes covering approximately 75% of the earth's surface contain critically important natural resources sustaining life and countless life cycles, while providing varying environmental, economic, social, and cultural services [1,2]. Such services are well documented and include the provision and storage of water supplies used for drinking and crop irrigation, power generation, oxygen production, nutrient supply and cycling, hosting important global food and pharmaceutical stocks, and climate regulation [3–5]. Furthermore, the use of water bodies for transporting commodities internationally using shipping fleets is significantly important for global commerce and trade. The provision of such services offered by aquatic biomes are often negatively impacted by continuing anthropogenic pressures [6,7].

Since the earliest times boats have served as transportation and played an important role in commerce and exploration [8]. To this day, boats continue to be an ever-present sighting throughout the world's waterways used for transportation, leisure, and sporting pursuits. Additionally, ships (distinguished by larger size, shape, and purpose compared with boats) historically, and currently,

play significant roles in ocean exploration, war-time conflicts, trade, fishing operations, migration, border security, infrastructure development and installations, as well as science/research-based efforts. The historical importance on boats and ships in today's society is reflected in the fact that many large and capital international cities owe their existence to the ease for vessel access as preferred sites of settlement. In fact, 22 of the 32 largest cities in the world are located on estuaries [9] with sheltered navigable waters and large associated infrastructure, including ever-expanding recreational marinas, commercial harbors, and trade ports highlighting the continuing importance to this day.

Boats vary in size and construction methods according to their intended purpose, available materials, or local traditions. Examples, include fishing boats which vary widely in style partly to match local conditions, pleasure craft used for recreational boating such as sailboats, ski boats, and pontoon boats, houseboats used for long-term residence or vacationing purposes, lighters and tugs used to transport cargo to and from other vessels and maneuvering assistance, respectively. Such boats are propelled either by manpower (e.g., oars or paddles), wind (e.g., sails), or engines (including fossil fuel and electric) and typically operate on inland waterways such as rivers and lakes, or in coastal waters. Ships, alternatively are larger ocean-going vessels capable of completing intercontinental travels and are designed for specific offshore purposes such as being capable of transferring high tonnages of commodities (e.g., tankers, dry and liquid cargo ships and floating production storage and offloading vessels) or large volumes of people (e.g., liners and cruisers), provide military services, or commercial fishing pursuits. Ship propulsion examples include steam turbines powered by fossil fuels or nuclear reaction, diesel engines, gas turbines, and stirling or steam engines.

Through the variety of services offered by boats and ships their worth to humanity and economic prosperity is apparent. Whether it is the provision of foods from small to large-scale fishing pursuits, transportation means for commercial and recreational purposes, leisure and sporting pursuits in more developed nations, and the global distribution of commodities. However, daily operations under normal boating and shipping practices, associated onshore activities (e.g., maintenance slips or ship demolition), or accidental discharges/occurrences (e.g., explosions, groundings, or loss of contents such as oil or merchant containers) all contribute to potential environmental impacts (e.g., [10–14]). Such related impacts affect aquatic biome services, most notably through decreased water quality, introduction of alien species, physical disturbance and/or destruction of habitat, flora and fauna, influencing system structure and function, in addition to atmospheric pollutant inputs [10,11,15–17]. Furthermore, impacts are either direct or indirect occurring over varying spatial and temporal scales, according to factors such as the size and density (e.g., cumulative impacts of clustered vessels), movement and specific activity-based operations of boats and ships, and environmental settings and conditions. For example, boating impacts on inland waters may be substantial since these water bodies are smaller and have more limited water exchange compared to coastal areas. Additionally, operational requirements and associated activities may give rise to particular impacts, which at times can be challenging to detect, measure, and predict. A firm understanding of these impacts in aquatic biomes is significantly important, both on immediate and long-term time scales.

Global population growth, increased trade, and forecasted increased use of recreational and tourism-based boating pursuits [11,18] will ensure that these boating- and shipping-related environmental impacts continue to increase without intervention and adoption of new technology (e.g., new antifouling agents, more energy efficient and environmentally friendly engines, environmentally friendly moorings (EFMs), etc.,). Accordingly, efforts are made to reduce these impacts in order to protect and preserve the environmental, economic, and cultural services aquatic biomes provide. The prevalence of boating- and shipping-related environmental impacts is detrimental to receiving environments and must be recognized and effectively managed to minimize the adverse effects, wherever possible, through the use of direct and indirect management tools, strategies and techniques (e.g., legislation, education, industry self-regulation, selective use of alternative technology, restrictions and management guidelines) to ensure the long-term sustainability of these environments.

However, the management and protection from boating- and shipping-related impacts are typically varied and challenging tasks given the complexities and interconnected processes at play within aquatic systems, in addition to impacts operating over varying spatial and temporal scales, varying user groups with competing needs and believed responsibilities, ambitions, economic interests, and potential exploitation of legal disparities, and inefficient coordination within and between different (nations) maritime zones [16,19,20]. Furthermore, given the transient and widespread nature of maritime zones, controlling their environmental impacts from numerous diffuse and mobile sources is much more difficult than managing discharges from smaller numbers of fixed-point sources, which often proves a resource and financially demanding assignment. Where there are large numbers of intermittent pollution sources, the cost of ensuring compliance through publicly funded patrols and enforcement is high, governments may only obtain political support for such an approach where the expected social and environmental benefits are equally significant [16]. Differences in boating-related environmental management measures have been shown to be associated principally with patterns in vessel size, which affects both the practical and regulatory requirements [16].

This work provides an up to date review of environmental impacts relating to boating and shipping to promote awareness and understanding of such impacts and implemented management instruments and considerations. For this review we consider boats and ships, regardless of their specific design, purpose, function, or associated group (i.e., industrial, commercial, recreational), providing an important resource for scientists, practitioners, and managers in identifying and prioritizing future management and research needs. First, we identify and discuss potential boating- and shipping-related environmental impacts, and hence the necessity for appropriate management measures. Second, an overview of a broad range of government- and industry-related instruments and measures for boating- and shipping-related environmental impact management is presented, in an Australian context. As noted by Byrnes et al. [16], as Australia is a large maritime continent covering approximately the same area as the United States of America (USA), the Australian-based management information presented herein will be generally applicable in similar settings worldwide and be generalizable, offering a microcosm for similar industries operating in many countries globally that can be used by resource managers to better target pollution regulations and programs for improved sustainability of aquatic biomes.

## 2. Boating- and Shipping-Related Environmental Impacts

Boating- and shipping-related environmental impacts are varied and include physical, chemical, and biotic factors, which although primarily impact aquatic biomes also include impacts to atmospheric and terrestrial zones. While oceanic regions are relatively less impacted by anthropogenic pressures, such pressures are exerting increased influences on coastal environments [21,22], and although much of this originates from land-based activities, considerable impacts are sourced from the operation and presence of boating and shipping operations and activities (at least cumulatively) through physical disturbances or the release of pollutants. Notable pollutants include antifouling agents, hydrocarbons, garbage, greenhouse gases (GHG), and sewage.

Physical disturbances comprise physical habitat and vegetation destruction through anchor damage, vessel groundings and wash, and fauna behavior modification (e.g., negatively impacting aquatic mammals, roosting birds, and fish) from vessel noise emissions and movements (e.g., [23–25]). However, attributing impacts to specific vessel-related activities can be difficult in systems used for multiple activities [15], and or with multiple anthropogenic inputs. Impacts may be either localized, such as physical disturbance from indiscriminate anchoring and mooring activities [26,27] or non-localized and dispersed, for example the drift and distribution of garbage objects or the leaching and dispersion of toxicants from antifoulant applications [12,28]. Additionally, physical, chemical, and biological impacts may occur either directly or indirectly, thus overlapping or transforming between physical, chemical, and biological impact categories. For example, initial physical disturbances that

directly damage the substrate and seagrasses, may then potentially lead to subsequent chemical and biological pressures [26,27,29].

An understanding of such impacts is important for the monitoring and reduction in terms of both immediate and longer-term effects for the sustainable integrity and provision of system services of aquatic biomes and their associated atmospheric and terrestrial zones. An account of the major boating- and shipping-related environmental impacts is provided below.

## 2.1. Physical Impacts and Influences

Boating and shipping operations result in physical and associated disturbances. Anchoring practices, traditional block and tackle moorings, and vessel groundings all disturb bottom substrates, seagrass meadows, and coral reefs. Furthermore, vessel presence, operational noise, and engine exhausts have also all been demonstrated to negatively impact the range of aquatic and bird species.

### 2.1.1. Anchoring and Mooring Activities

Traditional anchoring and mooring practices often represent a key disturbance to seagrass and other benthic habitats as a result of seabed scouring (chain drag) as tidal currents and wind conditions "swing" the buoy/moored vessel around the anchor point [14,27,30]. Additionally, direct damage from unrestricted anchoring poses a serious threat to reef communities with large anchors and chains able to pulverize coral colonies and smash reef structures [31,32]. Reported damage to coral reefs and seagrass meadows from recreational to large commercial vessels have long been reported globally in locations such as Australia [29,33], Brazil [34], Galapagos Islands [35], Philippines [36], and the USA and territories [37–39]. Habitat destruction and modifications of the physical and biological characters of the surrounding substrate raise concerns regarding, ecosystem integrity, with associated impacts including alterations to seagrass meadows (Figure 1), sediment resuspension thresholds, nutrient fluxes, organic matter content, sediment particle size distribution, and benthic faunal assemblages [26,27,29,32,40].

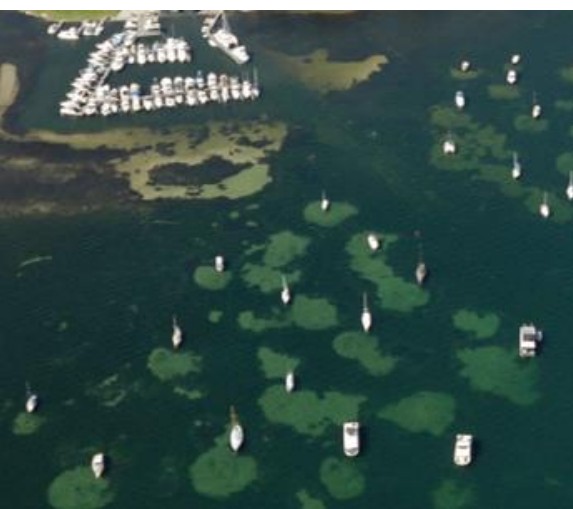

**Figure 1.** Seagrass meadow halos caused by traditional swing moorings, Marks Point, Lake Macquarie (Australia) (Reprinted from Aquatic Conservation: Marine and Freshwater Ecosystems, 28, Glasby and West [30], Dragging the chain: Quantifying continued losses of seagrasses from boat moorings, 383–394, Copyright (2018) with permission from John Wiley and Sons.).

As such, designated no anchor zones and the implementation of EFMs, designed to avoid scouring the seafloor, have been investigated in an effort to reduce the known environmental impacts placed upon the aquatic systems from anchoring and traditional buoy mooring approaches [30,40,41].

### 2.1.2. Groundings, Wrecks, and Abandonments

The physical damage caused by vessel groundings can be a source of significant disturbance and mortality to shallow coral reefs and soft bottom habitats [42,43]. Groundings of large vessels (i.e., >22.9 m in length [44]) can easily damage thousands of square meters of benthos [45], altering habitats, community structure and biomass, and potentially small-scale hydrodynamics. Physical damage to benthic habitat and organisms can be directly caused by the impact of vessel hulls, keels, or propellers or indirectly through the relocation of dislodged corals and the movement of sediments and rubble created during the initial impact (Figure 2a) [46]. The abrasive action of grounding can also result in scraped antifoulant paint containing the biocide being introduced to the environment [47,48].

Following the grounding incident of the 21,000-ton container ship *Bunga Teratai Satu* in the Great Barrier Reef (2000), injuries to hard and soft corals were observed both in the vicinity of the grounding site, and up to 250 m from the grounding, that were consistent with the symptoms expected from contact with antifoulants (paint samples retrieved from the grounding site confirmed that the hull was treated with a tri-*n*-butyltin (TBT)-based antifoulant) [47]. The recovery of such damage can be lengthy. For example, observations of impacted coral communities post a vessel grounding have shown limited signs of coral recovery more than three years after the initial impact with coral coverage still significantly higher in undamaged areas compared to the grounding scar [42]. Impacts of smaller ships and boats on coral reefs, often go unreported, but can represent a cumulatively larger source of coral mortality [42]. Works by Lutz [49] in the Florida Keys reported that 57.1% of shallow patch reefs sampled (*n* = 49) demonstrated evidence of boating impacts such as hull paint and scars on coral tissue, dislodged and fragmented corals, or crushed coral heads.

Additionally, abandoned/derelict vessels (Figure 2b) and (sub-surface) wreckages pose potential environmental impacts particularly in sensitive habitats such as coral reef, mangrove, or seagrass habitats. Such vessels represent potentially chronic source of pollution to the surrounding environment through delayed releases of oil, antifouling compounds, metals, and other toxic chemicals, while also potentially serving as a possible source of alien marine species [50–52]. Physical disturbances including direct damage to reefs and substrates and sediment erosion may also occur during periods of extreme wind speeds (e.g., storms, cyclones/typhoons).

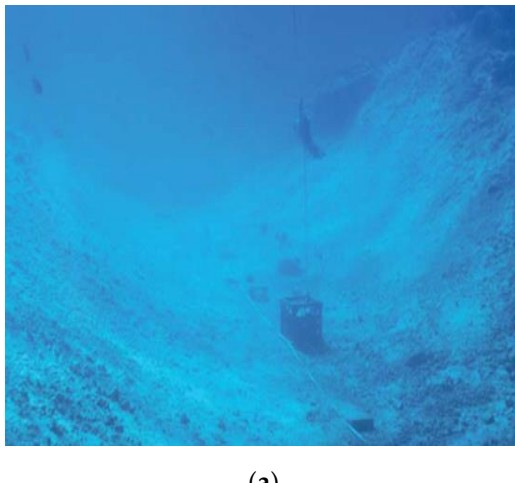　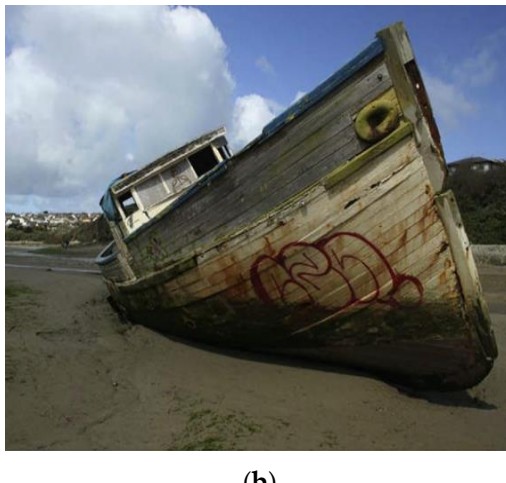

(**a**)　　　　　　　　　　　　　　　　　　　　　　　(**b**)

**Figure 2.** (**a**) Reef structure damage caused by the grounding of the ship *Bunga Teratai Satu*, Sudbury Reef, Great Barrier Reef (Australia), note survey assessment equipment in image (Reprinted from Spill Science & Technology Bulletin, 7, Marshall et al. [47], Grounded ship leaves TBT-based antifoulant on the Great Barrier Reef: An overview of the environmental response, 215–221, Copyright (2002), with permission from Elsevier.); (**b**) Abandoned boat, Gannel estuary (UK) (Reprinted from Marine Pollution Bulletin, 60, Turner [53], Marine pollution from antifouling paint particles, 159–171, Copyright (2010), with permission from Elsevier.).

### 2.1.3. Collisions and Disturbances (Fauna)

Vessel collisions represent a key hazard to a broad range of aquatic vertebrates (e.g., turtles, dolphin, whales, and manatees (e.g., [54–57]). Such hazards will presumably continue as a consequence of continued vessel traffic expansion. Additionally, smaller fauna is also at risk such as rays and swimming bird species [58,59]. Collisions often result in deep lacerations or anatomical injury from propeller-induced contact and/or internal injuries, or fatalities resulting from direct blows from vessel bows/hulls [55,56]. Vessel speed is one of the important factors that presumably determines the severity of a boat or ship collision but is also important regarding types of avoidance response and the efficacy of these strategies [59]. The use of jet propulsion systems, rather than traditional propellers on outboard motors, has been observed to reduce injuries and fatalities to sea turtles (and possibly other aquatic species) [54]. Vessel traffic disturbances potentially give rise to changes in behavioral traits in cetaceans, which may result in changes to habitat use, displacement and increased energy consumption [55], and survival rates or population size [60,61].

Increasing whale watching tour operators has prompted much research over recent decades (e.g., [55,56,62]). Such studies have established changes in whale and dolphin behavior in response to both vessel presence and noise emissions [63,64]. Additionally, Au and Green [65] report that the ensuing noise levels can potentially cause hearing damage to cetacean species [65]. Results from these studies also demonstrated a potential for habituation to tour boat operations, however, this is likely dependent on the species and specific experiences of an individual or group.

In addition, visual appearances and noise produced from spinning propellers have the ability to induce flight responses in bird species. Much of the early research investigating physical disturbances to fauna from vessel operations focused on reservoirs, lakes, and river systems utilized by waterfowl [66,67], which subsequently expanded to include both many types of water bodies and species (e.g., [68–70]). Of these studies, many concluded increased frequencies and durations of flushing responses (i.e., being frightened from cover) as a result of boating activities, which may also lead to reduced breeding success and negative survival consequences.

Because of the abovementioned factors, considerable research interest has emerged in the field (e.g., [71–74]), while they also support the implementation of appropriate mitigation and management strategies in regard to vessel operation while in close proximity to marine fauna [75].

### 2.1.4. Garbage and Debris

Garbage and debris sourced from vessels is a recognized key contributor to marine [76,77] and freshwater [78] pollution, largely originating from packaging materials discarded during repairs or catering, items or equipment dumped and discarded overboard, or lost (broken) parts of a vessel as a result of operating in heavy seas. Additionally, personal items (e.g., towels or clothing), maritime safety equipment, fishing gear (e.g., nylon line, metal sinkers), or military equipment [76,78,79] are also sources of aquatic pollution.

The harmful effects of garbage and debris on aquatic wildlife include ingestion and entanglement [80–82]. Modern plastics are characteristically a complex mix of polymers, residual monomers and additives with absorbed organic matter, chemical contaminants, and bacteria adding to their complexity [83]. The potential for harm to organisms arising from plastics is further increased by the transfer of contaminants to biota tissues as numerous plastic additives and chemicals can alter metabolic and reproductive endpoints [83].

The loss or abandonment of fishing equipment also poses a significant environmental impact threat through "ghost fishing," reportedly impacting a wide variety of species (Figure 3) [84–86]. Like vessel-sourced sewage pollution, garbage and debris poses visual aesthetic issues and regardless of any potential for environmental harm, most people believe that such pollution must be appropriately managed. The majority of garbage and debris constituent components are not readily biodegradable with their removal from the water column typically occurring via settlement and burial in the bottom

sediments or through shoreline accumulation, which may change the realm of the impacts but not necessarily the impacts themselves [83,87,88].

Garbage and debris may also act as transport vectors for alien species [80], relocating exotic taxa to otherwise unreachable frontiers, which may result in large environmental consequences. Additionally, lost cargo containers, particularly during adverse weather conditions, pose potential and varied environmental impacts through both physical damage to coral reefs and through loss of contents. Duhec et al. [89] report lost cargo containers are not uncommon and are a source of a wide variety of debris into the Indian Ocean.

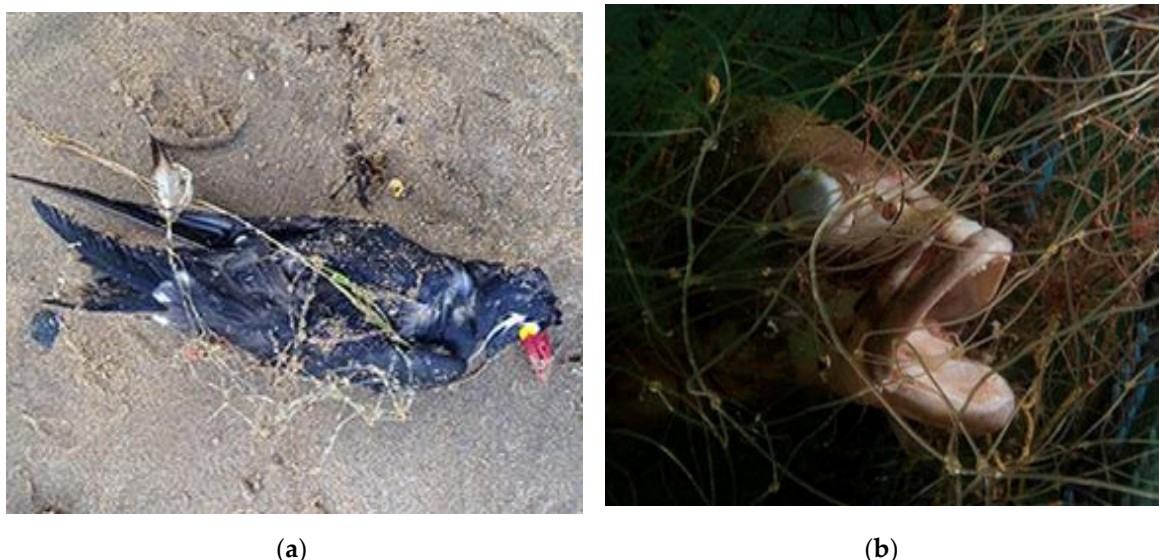

(**a**)                                              (**b**)

**Figure 3.** (**a**) Bird entangled in fishing net near Valparaiso (Chile), (Reprinted from Frontiers in Marine Science, 5, Thiel et al. [81], Impacts of marine plastic pollution from continental coasts to subtropical gyres—fish, seabirds, and other vertebrates in the SE Pacific, 238, Copyright (2018), frontiersin.org, (CC BY).). (**b**) Fish entangled in ghost net, Arvoredo Island (Brazil) (Reprinted from Perspectives in Ecology and Conservation, 17, Link et al. [90], Abandoned, lost or otherwise discarded fishing gear in Brazil: A review, 1–8, Copyright (2019), Elsevier (CC BY-NC-ND).).

2.1.5. Propeller Wash and Vessel Wake

Thrust fields from propellers and wash/depression waves from moving vessels potentially result in resuspending bottom sediments and physically impacting benthic and shoreline communities and habitat through bank and bed erosion [91–95]. Bottom resuspension may result from thrust fields from vessel propellers or solitary long waves that cause increased water velocities inducing resuspension. Factors influencing the anthropogenic resuspension from vessels includes size and speed of vessel, hull design, engine size, seabed sediment grain-sizes, degree of seabed cohesiveness, water depth, and under-keel clearance. The erosion and resuspension of bottom sediments have direct and indirect effects on system condition and function including increased suspended solids and correspondingly decreased light availability, changes in benthic community structure, potential smothering of biota, remobilization of toxicants and nutrients, and alterations to nutrient cycling processes and sediment transport [91,96–98].

Within protected aquatic environments where ambient wave energy is minimal, vessel wakes generated by recreational boats may also represent a substantial source of erosive energy directed at shorelines resulting in undercutting of banks, marsh loss, or degradation and disturbance to faunal communities [93,95,99]. The energy content of a produced wake impacting shorelines is influenced by vessel speed, hull-shape and displacement, vessel length and vessel proximity to the shoreline (i.e., distance travelled by the wake) [93,95,99,100]. Any resulting bank erosion or disruption to bank integrity is then dependent on factors such as shoreline type, slope, water levels and vegetation

community type, root depth and density [11,93,95]. Wake disturbance may also cause physical damage to oyster reefs and emergent and floating water plants leading to changes in the distribution of aquatic vegetation [94,95,99,101]. Wave heights of 0.3 m or less have been shown to erode vegetated shores or compromise marsh survival in protected aquatic environments [102,103].

Given the biological stress and morphological changes induced by vessel traffic in protected aquatic environments, adopted and proposed mitigation measures, and experimental investigations have been described. These include reductions in the navigation speed of vessels, increased distances between successive vessels entering narrow waterways, and limiting navigation during times of sufficiently deep tidal waters (e.g., [104]), and the effectiveness of natural breakwalls and oyster restoration structures in promoting sediment deposition and decreasing vegetation loss (e.g., [94]).

## 2.2. Chemical Impacts and Influences

Vessel operations, associated onshore activities, and accidental events all contribute to chemical environmental impacts with varying synergistic and cascading effects. The most critical chemical impacts result from operational and accidental discharges of hydrocarbons (i.e., fuels and oils), antifouling applications, and human waste (e.g., sewage effluent).

### 2.2.1. Antifouling Agents

Antifouling applications (paints) are applied on the outer layer of vessel hulls and other submerged parts of vessels to prevent the settlement and growth of aquatic organisms and in so doing maintain fuel efficient propulsion and/or proper maneuverability of vessels and limit potential for transfer of alien invasive species [105,106]. The mode of operation (passive flux), toxicity, and fate of antifoulants to marine fouling organisms pose environmental impacts to aquatic biota in the water column and benthic sediments, as such the understanding, distribution and fate of these agents has gained great attention [53,106–109]. Antifoulant applications used to impede the growth of fouling organisms include various impregnated biocides, such as chlorothalonil, copper, dichlofluanid, diuron, folpet, Irgarol 1051, Kathon 5287, maneb, TCMTB (2-(thiocyanomethyl thio)benzthiazole), TBT, zinc pyrithione (ZnPT), and zineb. The passive flux of these biocides from hull applications differs according to variables such as concentration, matrix (self-polishing copolymers, ablative, or epoxy-based formulations), operational factors, and ambient conditions. Within receiving waters and sediments variations in biocide concentrations have previously been reported to demonstrate the seasonal trends with maximum concentrations in high use areas coinciding with seasonal trends of peak boating [108,110–114].

Tri-*n*-butyltin used as early as the late 1960s [115], has been previously described as perhaps one of the most toxic substance deliberately released into aquatic environments [116], used either as the sole biocide or as an alternative to lead, arsenic, or organomercury boosters in copper-based antifouling paints [117]. The environmental impacts of TBT exposure include reductions in shell thickness, growth, and reproduction for a range of organisms (e.g., intersex in female gastropods whereby they develop male sexual characteristics as shown in Figure 4) [118–121]. As a result of their extensive environmental distribution and non-discriminatory biotoxicity, organotin biocide use in antifouling coatings has been restricted for ships <25 m in length in Organization for Economic Cooperation and Development member countries since 1988 [117], with a total ban on TBT use (and all other organotins) taking place internationally on 17th September 2008 [121].

Following these restrictions and a subsequent ban on TBT use, the number of booster biocide alternatives has increased, however many compounds (e.g., mercury, arsenic and DDT (dichlorodiphenyltrichloroethane)), have also been subsequently banned [122].

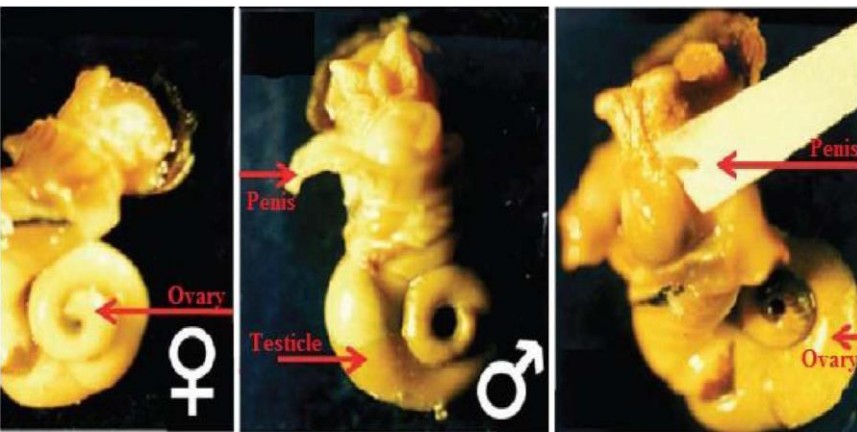

**Figure 4.** Development of penis in female gastropod *Ocenebra erinacea*, observed once in-water TBT concentrations exceed 1 ng L$^{-1}$ (Reprinted from Environmental Toxicology and Pharmacology, 57, Amara et al. [119], Antifouling processes and toxicity effects of antifouling paints on marine environment. A review, 115–130, Copyright (2018), with permission from Elsevier.).

An example booster biocide which garnered high use rates is Irgarol 1051 (2-methylthio-4-tert-butylamino-6-cyclopropylamino-s-triazine), which acts as an antifoulant by inhibiting photosynthesis [113,123,124]. While Irgarol 1051, at concentrations of as low as 136 ng L$^{-1}$ has been found to be toxic [112], concentrations of almost 1700 ng L$^{-1}$ has been reported in marine waters associated with vessel use [125]. Additionally, example concentrations in waters of Europe and Japan have been shown to range between 14 to 1571 ng L$^{-1}$ [112] and 10 to 262 ng L$^{-1}$ [126], respectively. With increased water column and benthic sediment concentrations, the potential for bioaccumulation of Irgarol 1051 has previously been investigated with factors ranging from 62 to 290 times for edible and non-edible fish reported, respectively [111]. Additionally, Irgarol 1051 has been reported to accumulate in macrophytes in freshwater environments by concentrations of almost 30,000 times that occurring in the surrounding waters [123]. Largely because of the potential harm to environment and human health risks posed by Irgarol 1051, its use has been restricted, or even banned in some countries for use on vessels <25 m in length (e.g., United Kingdom, Sweden and Denmark) [127]. Following the ban of Irgarol 1051 in the United Kingdom in 2001 on vessels <25 m in length, water and sediment sampling in southern England during 2004–2005 demonstrated significantly reduced concentrations in comparison to earlier studies (2000–2004), indicating that control measures by restricting the use of Irgarol 1051 had been effective in reducing its concentrations in coastal waters within the monitored region [107].

Currently, copper as an antifoulant ($Cu^{2+}$ or $Cu_2O$) is a widely utilized biocide that requires greater concentrations of "free" $Cu^{2+}$ (aq) to produce the necessary toxic effects [105,128] than does TBT. This type of copper antifoulant also has a greater potential for organic matter complexations than that of TBT [129,130]. However, elevated levels of copper still have deleterious effects on aquatic organisms and the increased input and accumulation of copper within aquatic biomes remain an environmental concern. Elevated concentrations of copper have been observed within sediments and waters of marinas and anchorages [12,114,131,132]. Field investigations into the release rates of copper from antifouling paints have reported that small to medium sized moored vessels emit $Cu^{2+}$ at rates of 8.2 μg cm$^{-2}$day$^{-1}$ [133] to 18–22 μg cm$^{-2}$day$^{-1}$ [134]. Smaller recreational and leisure boats are used in predominantly shallow nearshore areas where water exchange is limited and many species reproduce, thus posing presumably greater environmental impacts in comparison to larger vessels used in open oceanic water bodies with greater dilutive potential.

### 2.2.2. Gas Emissions

The principal source of vessel-related sources of GHGs is via the internal combustion engines used for propulsion or as a power source for onboard electrical equipment [135]. Other than smaller sailing vessels and dinghies, boats are typically driven by either outboard motors, inboard/outboard (such as stern drive units, for example) or inboard motors [136]. Typically, inboard/outboard and inboard motors are four-stroke (four-cycle) configuration, using either diesel or petrol as a fuel source, while outboard motors (particularly the smaller, lower powered units) are generally two-stroke (two-cycle) configured petrol engines that use a petrol/oil mix as a fuel source. As early as 1978, Chmura and Ross [137] determined that conventional two-stroke motors are, by their very design, less efficient than four-stroke motors, as not all the fuel mixture is burnt in the combustion process, which results in an unburnt fuel mix being expelled in the exhaust gases, which would otherwise be used to generate power. Regardless of the advances in two-stroke engine technology, the principal difference between engine types remains (e.g., a two-stroke engine can discharge up to 100 times more of its total fuel consumption into receiving waters than an equivalent four-stroke engine) (e.g., [138,139]). The high level hydrocarbon output from two-stroke engines is one of the key factors in regulations, driving their replacement [136,138]. Two-stroke engine emissions into receiving waters have also been identified as being far more toxic than those derived from four-stroke engines of equivalent power output [140]. The problems associated with two-stroke engine emissions have generated significant research interest (e.g., [136,141–144]).

The major anthropogenic GHGs from vessel operations, such as propulsion and the refrigeration of ship and container cargo, air-conditioned living quarters and occupied areas and refrigerated domestic food storage compartments, include nitrous oxide ($N_2O$), methane ($CH_4$), carbon dioxide ($CO_2$), and certain manufactured gases such as chlorofluorocarbons (CFCs), halocarbons, and their various substitutes. Such emissions potentially lead to decreased air and water quality (through atmospheric deposition potentially resulting in acidification, eutrophication), negative impacts to aquatic species through increased UVB radiation due to ozone layer depletion, and climate alteration [145–147]. Estimates regarding the shipping share of total air emissions to global anthropogenic emissions of $CO_2$, $SO_X$, $NO_X$ are; 3% [148], 4.5–13% [148,149], and 15% [148], respectively.

Strategies to reduce shipping emissions may include environmental evaluation schemes, greater public awareness, technological improvements, operational regulations such as for low-sulfur fuels and nitrogen oxide emission limits (e.g., IMO 2020, MARPOL Annex VI IMO) and stakeholder engagement [150].

### 2.2.3. Hydrocarbons

Boating and shipping present multiple pathways for the introduction of hydrocarbons into receiving aquatic environments. Sources include, structural vessel failure (Figure 5), leaks from poorly maintained engines, losses from fuel tanks/lines during operation or refueling, maintenance procedures, intentional and operational discharges, and from unburnt fuel in engine exhaust gases which quite often vent below waterline, particularly (and almost exclusively) so for outboard engines [15,151–153]. While bringing economic benefits to numerous countries through global supply chains marine oil shipping, although low in probability, witnesses oil spill accidents which potentially lead to serious pollution of receiving environments [152,154], often having devastating and long-lasting consequences [155–157].

Crude oil and refined petroleum products and their partial combustion products, contain toxic monocyclic (MAHs) and polycyclic aromatic hydrocarbons (PAHs) [158]. MAHs are typically of higher toxicity relative to PAHs, however, they generally evaporate much faster. PAHs have been described among the most mutagenic and toxic contaminates occurring in aquatic systems [159]. Furthermore, they are also known to produce long-term carcinogenic impacts on many organisms [160]. MAH and PAH toxicity levels relate to their octanol-water partition coefficient, with their overall impact being a result of the balance between their bioavailability and their toxicity once exposed [161]. The manner of action is narcotic, positively related to the soluble hydrocarbon concentration in the body tissues of the

organism [161]. Dissolved hydrocarbons are taken up by organisms directly from the water column by absorption through external surfaces and gills, as well as through the digestive tract.

An acutely toxic and (water-soluble) component of crude oil is the PAH naphthalene and its various derivatives [162]. PAHs typically exhibit low solubilities and high octanol-water partition coefficients [163]. Thus they are more likely to be found in association with benthic sediments and suspended solids in their receiving environments.

A study by Smith et al. [158] near Green Island in the Great Barrier Reef Marine Park (GBRMP) of Australia, reported increased PAH levels in areas where boating activities were known to occur. They observed that it was only the sediments adjacent to vessel moorings that exhibited low, but measurable levels of various PAHs in comparison to baseline concentrations at sites distant from boat moorings, which they speculated may have come from either fuel spillages or engine exhaust emissions [158]. For example, they found that benthic sediment concentrations for pyrene, anthracene, and benzo(a)pyrene ranged between <0.1–15, <0.06–1.0 and <0.004–4.3 $\mu$g kg$^{-1}$ dry wt, respectively, for three sites associated with boat moorings, while observed concentrations of <0.1–2.4, <0.06 and <0.004–0.8 $\mu$g kg$^{-1}$ dry wt, respectively, were reported for the 18 sites not immediately adjacent to boat moorings.

Similarly, Mastran et al. [160] reported that boating activity was a source of PAHs in water reservoirs in the USA, and that this was predominantly related to periods of increased boating activity. Additionally, they demonstrated localized impacts at marina sites by way of significant differences between total PAH sediment concentrations within marina versus non-marina sites [160]. These findings, and *Clean Air Act* amendments in the USA during the early 1990s, resulted in engine manufacturers implementing improved engine technology for better fuel-economies and reduced emissions, while lead and benzene were also prohibited as fuel additives to aid in the overall reduction of emissions.

Estimations that petroleum products are potentially lethal to adult stages of various aquatic organisms (e.g., fish, mollusk, crustacean and flora species) at concentrations ranging from 1–100 mg L$^{-1}$ [164], while concentrations of 0.01–1 mg L$^{-1}$ may be lethal to their larval and juvenile stages. Furthermore, concentrations as low as 0.001 mg L$^{-1}$ may result in sublethal effects to the behavior and reproduction of various marine organisms [158]. However, notwithstanding the immediate toxicity of aromatic hydrocarbons, the physical impacts of oil spills can cause significant environmental harm, dependent upon the volume of oil spilled and conditions of the receiving environment [165–167]. Generally observed physical effects to marine biota include impairment of respiratory surfaces (i.e., fish gills and mangrove pneumatophores), feeding apparatus (e.g., in oysters and mussels), and damage to protective layers (e.g., fur and feathers), or the entire smothering of whole benthic or shoreline habitats [168]. Ecological impacts have been estimated to occur at, or above, 10 g m$^{-2}$ (thickness of ~10 $\mu$m) on the water surface [169] as this level of oiling has been recorded to fatally impact bird species through the adhesion of oil to feathers, exposing individuals to (secondary) effects such as hypothermia.

An additional class of hydrocarbons associated with internal combustion engines are particular fuel additives (oxygenates) (e.g., methyl tert-butyl ether (MTBE)), that are designed to help improve combustion efficiency and ultimately air quality. Interestingly, increased concentrations of these compounds have reportedly been found in connection with recreational boating activities [138,139,170]—such as Dinerman et al. [139], who estimated that the average annual load of MTBE from boating activities into Lake Kinneret (Israel) was approximately 4430 kg yr$^{-1}$ and that 2.3% of this value was attributable to boats with four-stroke engines, while 61.4% and 36.3% was attributable to boats and jet-skis with two-stroke engines, respectively.

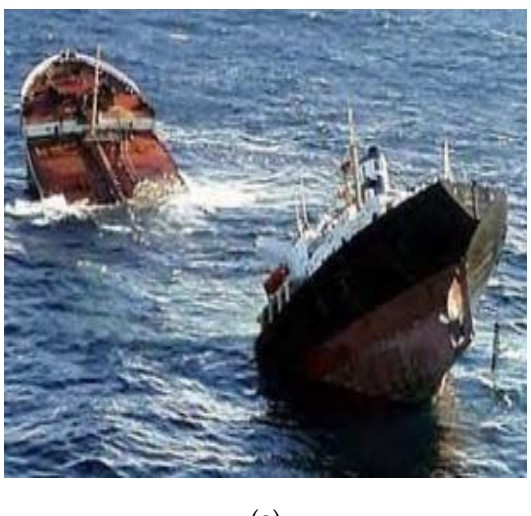 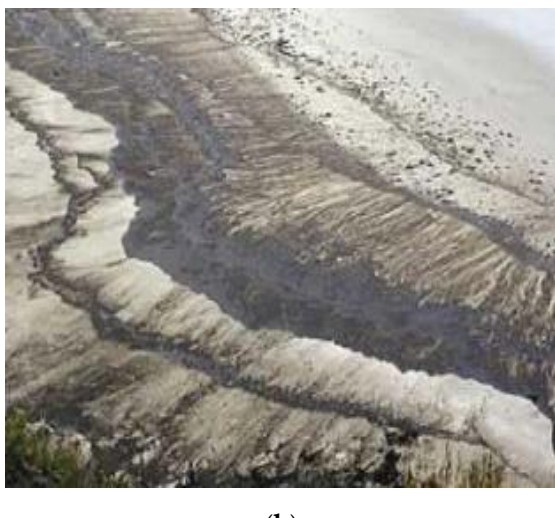

(**a**)                                    (**b**)

**Figure 5.** (**a**) Catastrophic failure of *Prestige* tanker 19 November 2002, off Finisterre coast (Spain) (Reprinted from Marine Pollution Bulletin, 53, Albaigés et al. [171], The Prestige oil spill: A scientific response, 205–207, Copyright (2006), with permission from Elsevier.). (**b**) Oiled beach on Galacia coast (Spain) resulting from failure of *Prestige* tanker (Reprinted from Marine Pollution Bulletin, 53, Albaigés et al. [171], The Prestige oil spill: A scientific response, 205–207, Copyright (2006), with permission from Elsevier.).

### 2.2.4. Maintenance and Ship Breaking

For both vessel safety and environmental impact concerns, boats and ships typically require routine upkeep and maintenance to operate to their intended design specifications. Depending on the size and required maintenance such undertakings occur typically either in marina waters, or slipways and hardstands or in dry docks for larger ships. Furthermore, maintenance requirements often involve tasks associated with structural, electrical, or mechanical maintenance procedures. Maintenance and upkeep procedures without correct waste containment, management and disposal plans, and procedures can potentially provide a source of varied contaminants, such as spent engine fluids, waste hydrocarbons and used ethylene glycol (anti-freeze), waste solvents from parts-cleaning operations; detergents; paints; vessel scrapings and dust; metals from worn parts and replacement batteries and acids. Such potential pollutants can be toxic to aquatic organisms or have legacy effects raising environmental concerns at maintenance stations/areas [172–175].

At the completion of a vessel's life cycle, when refitting and maintenance becomes uneconomical, large ships often undergo ship breaking or ship demolition. The process of dismantling an obsolete ship's structure and components for scrapping (recycling) or disposal, is often conducted at piers, dry docks, or beaches/shorelines [Figure 6] and includes a wide range of activities, with associated environmental impacts including the discharge of harmful and persistent pollutants.

Such pollutants include asbestos, polycyclic biphenyls (PCBs), dioxins, sewage, total hydrocarbons and PAHs, organo-tins (and other antifouling biocides), bacterial contaminants, and heavy metals, which all pose risk of environmental impact [176–180]. Heavy metals, including sources from electrical wiring and systems (e.g., Cu, Pb, Hg), batteries (e.g., Pb, Ni and Cd), coatings, and paints (e.g., Cu, Zn, Cl, TBT) have been demonstrated to accumulate to greatly elevated concentrations within coastal waters and beach sediments where ship-breaking activities are practiced. For example, many studies have identified significantly elevated concentrations of metal concentrations, including in airborne particulate matter, nearshore and ground waters, and sediments, in ship breaking zones in comparison to nearby reference zones in countries such as Bangladesh, India, and Turkey [179,181–183]. The severity of environmental impacts from breaking practices is reportedly varied according to the size and function of the ships [179,184]. Although ship breaking practices have many environmental

issues involved, it is a great source of importance to many economies, e.g., in Bangladesh, India, Pakistan, and Turkey [179,184], often whom have lax or no environmental laws nor enforcement of laws [184].

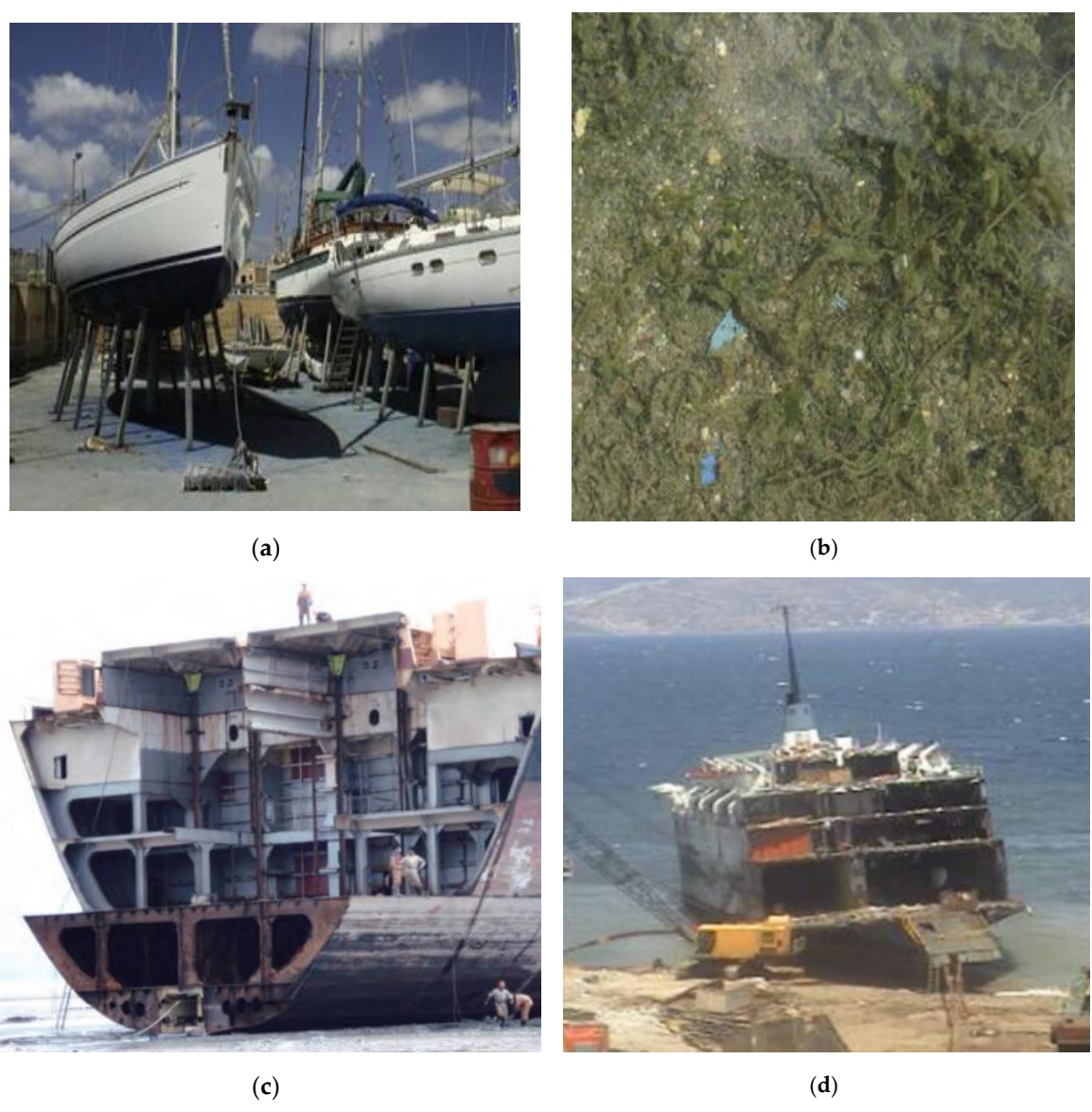

**Figure 6.** (**a**) Dust under vessel from antifouling paint during maintenance (Malta) (Reprinted from Marine Pollution Bulletin, 60, Turner [53], Marine pollution from antifouling paint particles, 159–171, Copyright (2010), with permission from Elsevier.). (**b**) Chips of antifouling paint on seabed adjacent vessel maintenance stands (Malta) (Reprinted from Marine Pollution Bulletin, 60, Turner [53], Marine pollution from antifouling paint particles, 159–171, Copyright (2010), with permission from Elsevier.); (**c**) Ship breaking activity on a shoreline in Chittagong (Bangladesh) (Reprinted from Sarraf et al. [180], The ship breaking and recycling industry in Bangladesh and Pakistan, Copyright (2010), openknowledge.worldbank.org (CC BY).). (**d**) Ship breaking activity on a shoreline in Aliağa (Turkey) (Reprinted from Journal of Cleaner Production, 16, Neşer et al. [182], The shipbreaking industry in Turkey: Environmental, safety and health issues, 350–358, Copyright (2008), with permission from Elsevier.).

2.2.5. Sewage

Sources of sewage from boats and ships include occupants defecating or urinating directly into the waterbody, toilets being flushing directly into the receiving waters or discharges from on board holding tanks or sewage treatment systems [18,185]. Sewage contains elevated nutrient concentrations (e.g., both nitrogen (N) and phosphorus (P)), in addition to fecal bacteria, which may be at levels several orders of magnitude above background concentrations of receiving environments [186]. Increased P and N loadings resulting from sewage discharges may lead to significant alterations in the structure and function of aquatic systems that rely on low ambient nutrient concentrations (e.g., corals and seagrasses) [187,188] which, in extreme conditions, can be replaced by algal assemblages [189].

Boating- and shipping-related sewage impacts would generally be considered in a cumulative sense rather than in isolation (i.e., in combination with land-based activities). This source of sewage is considered to be most problematic within enclosed inland waters and/or semi-enclosed coastal waters with minimal flushing [190]. Additional scenarios where vessel-sourced sewage discharges are problematic, include those that occur within high conservation areas (e.g., marine parks), areas continuously receiving high volumes of vessel traffic with relatively large numbers of people on board, or where activities involving in-water primary human contact occurs (e.g., swimming, fishing, or diving activities). The size, depth, and (tidal) flushing volume of the receiving environments, in combination with vessel use characteristics must all be considered when determining the sewage capacity of any given waterbody [137].

Vessel-sourced sewage discharges can be either at low continuous rates (e.g., direct releases from onboard toilets) or via large peaks (e.g., pump outs of holding tanks while at sea). The volume of sewage discharged by a particular vessel is a product of passenger numbers and the on board sewage management equipment rather than overall vessel size. Owing to the environmental impact unregulated sewage discharges may cause to receiving environments, boat and ship-sourced discharges of sewage has attracted much interest (e.g., [18,191–194]) with some preliminary attempts being made to estimate the potential volumes of discharged sewage and associated parameters being made [18,195].

Furthermore, given the potential environmental impacts and effects of sewage discharges, many countries enforce requirements by law to "pump out" untreated sewage from their holding tanks into land-based sewage treatment infrastructure. In many instances to achieve this, vessels may either dock at and use onshore stationary sewage reception facilities or a pump-out truck, or alternatively be serviced by pump-out boats, which navigate within and between marinas to collect sewage from other vessels.

2.2.6. Trace Metals

In addition to the abovementioned sources of metals (e.g., debris, wrecks, antifoulant applications and ship breaking), boating and shipping operations also act as a source of trace metals into receiving aquatic environments through other (operational) pathways such as; ballast water discharge, corrosion and use of sacrificial anodes, mechanical abrasion and engine exhausts [196–198]. The variable array and persistence of such sources ensures potential metal accumulation in the sediments of lakes, estuaries, and coastal waters, most particularly in high traffic, high density settings within protected waters (reduced flushing).

There are several trace metals that are vital biological elements, however, they can also be toxic to organisms above given threshold concentrations [199]. Even elevated concentrations of metals such as Cu, zinc (Zn), aluminum (Al), and lead (Pb), that are below their toxic thresholds values, can still cause damage to the physiological processes of certain organisms, chiefly their respiratory organs (i.e., gills) and central nervous systems [200].

Long-term inputs of metals arising from vessel sources may have long-lasting effects on the receiving environment within established marinas, harbors, and ports. A study by Dobaradaran et al. [198], investigating vessels entering Bushehr Port (Persian Gulf, western Iran), observed concentration levels of Cu and iron (Fe) in all samples of the ballast water were higher

compared with the coastal waters of Bushehr Port. Additionally, in the case of cadmium (Cd), 76.47% of sampled ballast water from entering vessels had higher concentration levels compared with the coastal waters of Bushehr Port. These findings illustrate the potential for long-term metal accumulation from ballast water as an example.

*2.3. Biological Impacts and Influences*

Important biological impacts arising from vessel operations is the potential continued introduction and secondary spread of alien (non-native) species into receiving aquatic biomes. Additionally, vessel operations and onshore associated infrastructure also alter water column light conditions impacting biological processes and potentially system condition.

2.3.1. Alien (Non-Indigenous) Species

Boating and shipping acting as vectors for alien (non-indigenous) species pose significant environmental (and economic) threats to freshwater, estuarine, and marine systems [11,201,202]. The introduction and spread of alien species beyond their native range is an environmental impact issue worldwide. The relocation of such species may include, fouling on hulls of recreational vessels, foreign sourced ballast water, internal water systems, vessel ropes, chains, vessel cavities, and sediments [11,201,203–206], in addition to fouled trailers, propellers, fishing gear, and bait buckets, which are of particular relevance in freshwater systems [207,208]. Smaller recreational vessels that are often moored within or near international commercial ports are known secondary vectors via hull fouling [201]. Such smaller boats may access shallow environments not accessible by large commercial ships [201].

Alien species invasion involves initial arrival and dispersion (uptake, transport, and release), establishment of species requiring survival and reproduction, and species spread [209]. Simard et al. [201] reports invasions of alien species success is influenced by colonization pressures (i.e., total number of species released in receiving region) and propagule pressure (i.e., number of discrete introduction events and number of individuals released in an introduction event). High connectivity between uninvaded and invaded regions enhances invasion risk [201].

Alien species sourced from vessel activity include diverse taxonomic groups, such as acidians, algae, bivalves, bryozoans [201,210–212]. Well-document example cases include the green mussel, *Perna viridis* (Linnaeus, 1758) [213], the black-striped mussel, *Mytilopsis sallei* (Récluz, 1849) [214], and northern Pacific seastar, *Asterias amurensis* [215].

Although it is possible to eradicate alien spices following establishment [216,217], this outcome is typically rare, and most species persist in their new range following establishment [218]. As such, preventing introduction of alien species beyond their native territory is of vital importance in addressing this boating- and shipping-related environmental impact.

2.3.2. Light Conditions

Boating, shipping, and associated marina and port infrastructure operations contribute to artificial light pollution (Figure 7), altering natural colors, cycles, and intensities of nighttime light [219,220]. Light pollution is globally widespread in marine environments [221].

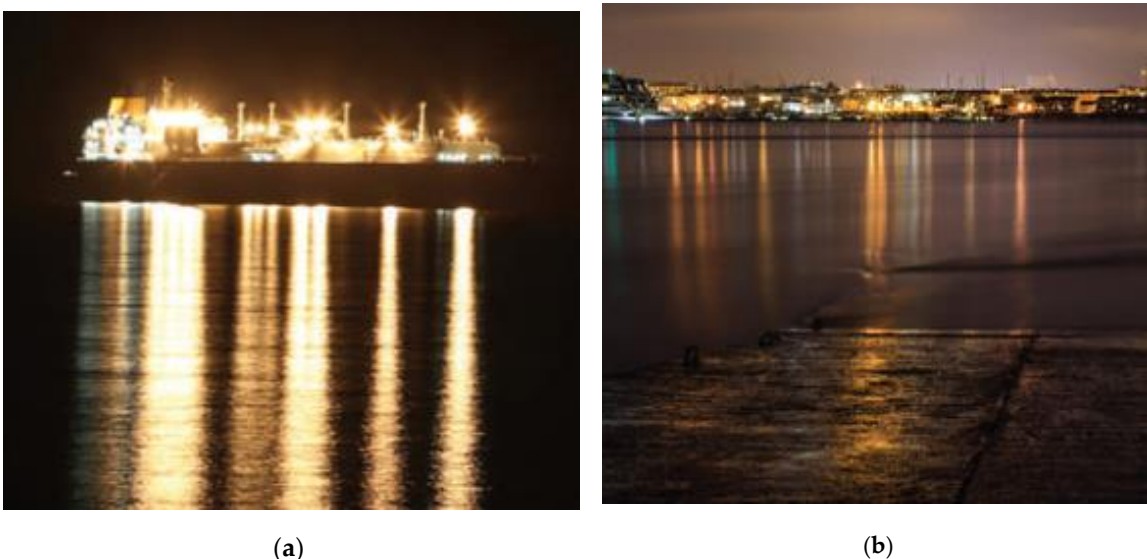

| (**a**) | (**b**) |

**Figure 7.** (**a**) Artificial light from ship, Falmouth Bay (UK) (Reprinted from Frontiers in Ecology and Table 12. Davies et al. [221], The nature, extent, and ecological implications of marine light pollution, 347–355, Copyright (2014), with permission from John Wiley and Sons.). (**b**) Lighting from harbor and infrastructure, Tamar estuary (UK) (Reprinted from Frontiers in Ecology and the Environment, 12, Davies et al. [221], The nature, extent, and ecological implications of marine light pollution, 347–355, Copyright (2014), with permission from John Wiley and Sons.).

Known and potential impacts from changes to ambient light regimes include: (i) Hindrances to navigation, migration, and communication; (ii) suppression of zooplankton diel vertical migration by artificial skyglow; (iii) aggregating fish under lighting leading to intensified predation; (iv) nighttime bird strikes on illuminated vessels; (v) alterations in foraging behavior in wading birds; (vi) altered recruitment and site selection of invertebrate larvae; (vii) stressors to coral communities and changes to timing of coral spawning events; and (viii) displacement of nesting sea turtles and disorientation and increased predation of hatchlings [219,221–225].

In contrast, the same boating, shipping and associated marina and port infrastructure and operations can also have an impact on aquatic environments by reducing available light for the water column and shallow surface sediments. Examples include long-term moored vessels, docks, wharfs, jetties, and pontoon moorings. Light and shading are known to reduce photosynthesis, growth, and depth distribution of seagrasses [226,227], alter water column and benthic primary production, metabolism and nutrient dynamics [228,229] and benthic community structure and function [230,231]. Studies documenting the effects of shading by dock structures on seagrasses include those by Loflin [232] and Shafer [233].

This section focused on the key environmental impacts arising from boating and shipping activities and related infrastructure, which require appropriate management to limit harm to the environment and ensure the sustainable use of aquatic resources (Table 1). In order to appropriately manage the above-mentioned impacts, there exists a range of available options, measures, and strategies. An outline of relevant government- and industry-related instrument for boating- and shipping-related environmental impact management, including examples of various management measures, are presented in the following section.

**Table 1.** Key pressures of boating and shipping operations (and associated infrastructure and activities) on water and air quality, inhabiting biota and related environmental impacts in aquatic biomes and example protection and/or management measures.

| System Element | Example Driver | Key Pressures | Example Protection and/or Management Measures |
|---|---|---|---|
| Physical changes to bottom substrate and habitats | Anchoring and mooring | − Physical disturbance to benthic habitat<br>− Loss of habitat (e.g., seagrass and corals)<br>− Alteration of sediment condition and resuspension thresholds<br>− Alteration to biogeochemical processes<br>− Alteration of community biomass and structure | Designated anchorages, adoption of environmentally friendly moorings, zoning plans, regulations, community education, restoration |
| | Groundings/wrecks/abandonments | − Physical disturbance to benthic habitat<br>− Loss of habitat (e.g., seagrass and corals) and substrate scouring<br>− Alteration of community biomass and structure<br>− Small-scale hydrodynamic change<br>− Chronic source of pollution | Pilotage services, artificial intelligence navigation, vessel crew education, training and awareness, propeller guards, imposed speed restrictions, governance and regulations, vessel removal (e.g., Queensland's "War on Wrecks" initiative [234]). |
| | Dredging (channel maintenance) and canal development | − Physical disturbance to benthic habitat and aquatic fauna<br>− Change to tidal flows hydrodynamic characteristics (i.e., flow velocities, tidal prism, patterns) and sediment transport processes (e.g., erosion and deposition patterns)<br>− Increased loading of suspended solids, nutrients, organic matter<br>− Alteration to biogeochemical processes<br>− Alteration to sediment transport processes<br>− Disturbance to shorebird nesting, roosting foraging sites at sediment disposal sites | Flexible dredge management plans, silt curtains, turtle exclusion devices (TEDs), compliance regulations and monitoring (dredge plume, aquatic fauna, shorebirds) |

**Table 1.** *Cont.*

| System Element | Example Driver | Key Pressures | Example Protection and/or Management Measures |
|---|---|---|---|
| | Thrust fields from vessel propellers or hull displacement waves | − Increased loading of suspended solids, nutrients, organic matter<br>− Increased light attenuation<br>− Increased shoreline erosion<br>− Smothering of benthic biota<br>− Remobilization of toxicants and nutrients<br>− Alterations to nutrient cycling processes | Implemented speed limits, sufficient depth and width of navigation channels, operator education and awareness |
| | Light | − Alteration to sediment primary production, metabolism and nutrient dynamics<br>− Alteration of benthic community structure and function | Shoreline and overwater construction guidelines, flexible designs |
| Physical changes and anthropogenic pressures to foreshore | Foreshore development (e.g., harbor, marina and port infrastructure and associated services) and increased impervious surfaces | − Physical habitat disturbance<br>− Elevated peak runoff and overall volume with a reduction in the timing and peak flows from rainfall events<br>− Change in quality of stormwater<br>− Change to tidal flows hydrodynamic characteristics (i.e., flow velocities, tidal prism, patterns)<br>− Increased potential debris and pollutant loads (e.g., suspended solids, metals, nutrients, hydrocarbons)<br>− Alteration of productivity and biogeochemical processes | Restoration of riparian habitats, constructed wetlands, bioretention, rainwater harvesting, grass channels, vegetated swales and strips, infiltration basins and trenches, porous pavements, soft engineering strategies, sediment traps, gross pollutant traps, grate covers, booms, community education |
| | Vessel wake | − Foreshore erosion or disruption to bank integrity<br>− Morphological changes<br>− Physical damage and changes to vegetation communities | Implemented speed limits, revegetation of banks, adaptive designs to minimize erosion processes, erosion monitoring, operator education and training, signage and aids to navigation, optimized navigational channel placement (where possible/practical) |
| | Ship breaking | − Source of hazardous and noxious substances (e.g., asbestos and polychlorinated biphenyls)<br>− Chronic source of pollutants<br>− Decrease in water and sediment quality | Regulated industry practices, governance, industry education, pollutant reduction measures, appropriate recycling and storage facilities, correct handling and disposal of materials |

**Table 1.** *Cont.*

| System Element | Example Driver | Key Pressures | Example Protection and/or Management Measures |
|---|---|---|---|
| Alteration to physico-chemical water column properties and aquatic biota | Antifouling application | − Leaching of toxicants (e.g., TBT, arsenic, copper, lead, zinc, biocides) decreasing water and sediment quality<br>− High toxicity to non-target marine biota<br>− Adverse effects on seagrasses and benthic biota<br>− Significant bioaccumulation and food web impacts | Governing regulations (e.g., *Protection of the Sea (Harmful Anti-fouling Systems) Act 2006* (Cth)), restrictions and total bans (select antifoulants), development and application of alternative "green" antifouling biocides, alternative antifouling measures (e.g., removing vessels from water when not in use where possible/practical, physical removal such as through hull scrubbing) |
| | Operational and accidental discharges (ballast and bilge water, hydrocarbons and sewage (treated and untreated)) | − Increased pollutant loads (e.g., total suspended solids, nutrients, fecal bacteria, hydrocarbons)<br>− Broadscale from DNA-damage to altering community structure<br>− Decrease in water and sediment quality particularly in enclosed inland waters and semi-enclosed coastal waters where flushing is minimal (e.g., elevated nutrients, bacteria and decreased dissolved oxygen)<br>− Potential introduction of alien (non-indigenous) species | Laws, regulations and restrictions governing operational discharges (e.g., IMO International Convention for the Control and Management of Ships' Ballast Water and Sediments), use of oil drip trays, absorbent materials, engine servicing and maintenance, care and vigilance during re-fueling procedures, improved engine technology, oily water/waste collection and appropriate storage and onshore disposal, appropriate spill response equipment, environmental plans, contingency and preparedness plans, oil spill trajectory modelling to aid response measures, mitigation measures (e.g., booms, dispersants, skimmers), sewage holding tanks and pump-out facilities, sewage treatment systems/marine sanitation devices |
| | Fauna collisions/disturbance | − Deep lacerations or anatomical injury from propellers<br>− Internal injuries or fatalities resulting from direct blows from vessel bows/hulls<br>− Disturbances causing alterations in behavioral traits (e.g., habitat use, displacement and increased energy consumption)<br>− Prolonged disturbances potentially alter survival rates or population size | Operational guidelines and regulations to maintain buffer between observed fauna (e.g., whales) and vessel, zoning plans, exclusion zones, speed limits/reductions, use of jet propulsion systems (in place of traditional propulsion by engines fitted with propellers), operational permits, operator education and training |

**Table 1.** *Cont.*

| System Element | Example Driver | Key Pressures | Example Protection and/or Management Measures |
|---|---|---|---|
| | Garbage and debris | – Garbage and debris ingestion and entanglement<br>– Leaching of adsorbed chemicals<br>– Bioaccumulation of microplastics/chemicals<br>– Introduction of alien species rafting on debris | Education campaigns and community awareness programs, behavioral and operational changes, ensuring that on board items are stowed safely and/or securely fastened, increased use of biodegradable packaging, appropriate garbage collection facility practices, provision of clearly labelled garbage bins and ashtrays, operator instructions regarding appropriate garbage management practices on board (particularly use of garbage bins and ashtrays) |
| | Vessel transits | – Physical disturbance to benthic and shoreline habitat<br>– Source of trace metals, herbicides and hydrocarbons, coolants, paints, degreasing agents, etc.<br>– Source of garbage and debris (e.g., lost or abandoned fishing equipment, cargo containers)<br>– Operational discharges (see above)<br>– Artificial light<br>– Potential introduction of alien (non-indigenous) species<br>– Elevated noise and vibration from engines, propellers, thrusters, power generators, machinery (e.g., cable/pipe laying vessel operations) or survey vessels (seismic) adversely impact communication, feeding, reproduction and navigation of aquatic biota and bird species | Laws, regulations and restrictions governing operational discharges (e.g., MARPOL), stakeholder education, environmental management plans, rubbish minimization initiatives, mitigation measures including: reduction in vessel speed, hull maintenance, optimization and potential redesign of vessel propulsion, alteration of navigation routes, zoning plans, sewage holding tanks and pump-out facilities, sewage treatment systems/marine sanitation devices, alternate environmentally friendly antifouling coatings, vessel use/speed restrictions, advanced wastewater treatment facilities |
| | Vessel maintenance and repair | – Source of trace metals, herbicides and hydrocarbons, coolants, paints, degreasing agents, etc. | Maintenance and repairs to be conducted out of the water (e.g., dry docks, marina hard stands) at appropriately designed and operated facilities that ensures that all wastes are contained, collected and contaminants appropriately disposed |
| Alteration to atmospheric conditions and processes | Exhaust gas, nitrogen and sulfur oxides, carbon monoxide and dioxide, volatile organic compounds and particulate emissions | – Decrease in air quality (chemical and particulate)<br>– Decrease in water quality through atmospheric deposition (e.g., potential acidification, eutrophication)<br>– Climate alteration<br>– Acid rain | Technological improvements, international regulations for low-sulfur fuels and nitrogen oxide emission limits (e.g., IMO MARPOL Annex VI), alternative energy sources, scrubbers, zero emission berth standard in ports by use of shore-side electricity |

**Table 1.** *Cont.*

| System Element | Example Driver | Key Pressures | Example Protection and/or Management Measures |
|---|---|---|---|
| | Refrigerate gas emissions (ozone-depleting substances: CFC/HCFC) | − Increased UVB radiation due to ozone layer depletion negatively effecting aquatic species<br>− Potential change to aquatic community structure | Phase out use of halons and ozone-depleting CFCs through design limitation of new vessels, emission limits (e.g., IMO MARPOL Annex VI) phasing out of halons and ozone-depleting CFCs |
| | Light | − Hindrances to spp. navigation, migration and communication<br>− Suppression of zooplankton diel vertical migration<br>− Bird strikes on illuminated vessels<br>− Alterations in spp. behavior and increased predation<br>− Stressors to coral communities<br>− Disrupted circadian rhythms | Light pollution guidelines (e.g., Australian Government's "National Light Pollution Guidelines for Wildlife" [235]), enforced/voluntary light management practices, including: reduction and/or cover of light sources on dark nights (especially when visibility is low), avoiding high-risk areas for bird strikes whenever possible on nights with poor visibility, reduction of illumination sources (e.g., directional lighting or light baffling), avoid nesting/breeding sites where possible/practical, operator education and training |

CFC = chlorofluorocarbon; HCFC = hydrochlorofluorocarbons.

### 3. Environmental Impact Management

The major boating- and shipping-associated impacts outlined above highlight the need for their appropriate management to ensure that aquatic biomes are protected, used sustainably, and continue to provide the best possible environmental (and human health) outcomes. The following section outlines the example measures that can be used to sustainably manage these boating- and shipping-related environmental impacts.

*3.1. Background*

A range of measures are available to help manage the potential environmental impacts stemming from human activities, such as: (i) laws/legislation, (ii) environmental management guidelines, (iii) industry codes of practice/conduct, and (iv) general environmental type information developed and distributed by various non-government conservation and/or environmental groups (there is also the potential for the application of simple "common sense" measures in many cases).

Such measures are generally discussed under the following three broad categories or instrument types as follows: (i) Direct regulatory instruments—aimed at directly influencing behavior through the use of laws, including permits, standard-setting and zoning or planning, that directly control or restrict environmentally damaging activities (also known as "command and control" instruments, such as traditional legislation, regulations, etc.,); (ii) informational and motivational instruments—aimed at shifting individual or community preference more toward conservation and inform or educate people regarding relationships between their activities and the environment (also known as "persuasive instruments" as they attempt to "persuade" individuals, groups, organizations, etc., to foster greater environmental responsibility through education, training, information, voluntary agreements, etc.,); and (iii) economic instruments—based on the principle that the "polluter pays" (e.g., environmental taxes, charges, and fees), that put a price on environmentally damaging behavior or payments for environmental services and ecological fiscal transfers that reward conservation-enhancing behavior (e.g., [236]).

Government authorities have tended to focus on the direct regulatory approach, typically through the use of legislative regulations and policing (e.g., approvals, fines, licensing, etc.,) in an attempt to minimize or avoid the adverse environmental impacts generally associated with industrial and economic growth. Historically, this approach has been relatively effective in an overall reduction of environmental destruction and pollution, particularly where willful or careless actions, gross negligence, or widespread malpractice have been evident. However, they have also lacked flexibility and, if designed poorly have been expensive to administer [237–239]. Following increased global development and technology, direct regulatory instruments have received considerable criticism, often from affected industries. In response, academic, government, industrial, and international agencies proposed and developed a varying array of management instruments designed to complement or, in some cases, substitute the traditional style of regulation. Such supplementary management measures provide various levels of industry-based participation and self-regulation [240,241], which may be broadly grouped, based on their principle action as either "organizational and operational management," "education," or "economic"-based instruments.

Examples of organizational and operational management-based instruments include industry self-regulation schemes, such as the ISO 14000 family of standards for Environmental Management Systems (EMSs) established by the International Organization for Standardization (ISO), providing a structure for businesses to manage their environmental impacts [242], the American Chemistry Council's Responsible Care® initiative [243], and self-regulatory programs established in the recreational dive industry, by various training and accreditation organizations like PADI (Professional Association of Dive Instructors) [244] and NAUI (National Association of Underwater Instructors) [245], which aim to ensure recreational divers are appropriately trained and assessed as competent prior to undertaking recreational diving activities. Second, environmental management instruments designed to provide consumers (or other relevant stakeholders) with education and information including Community

Right-to-Know (CRTK) schemes, public corporate environmental reporting, pollutant inventories, environmental award systems, and green labels (and/or other product certification) [239]. Finally, economic instruments are similarly extensive and include varying options including market creation (i.e., tradeable pollution rights), property rights (e.g., water), fiscal instruments, charge systems (i.e., green taxes), performance bonds, low interest credits, special funds, etc., [239].

*3.2. Direct Regulatory Instruments*

Boating- and shipping-associated impacts have attracted much attention over the past half-a-century. Following multiple oil spills associated with major tanker incidents, the International Maritime Organization (IMO) in 1973 organized a conference of government representatives with the overall aim of establishing a regulatory system for all pollutant classes associated with international shipping. Ultimately, this led to the International Convention for the Prevention of Pollution from Ships (MARPOL 73/78), which was ratified on the 2nd October 1983. MARPOL contains six technical annexes for the regulation of prevention of pollution from ships (Table 2).

**Table 2.** The six annexes of MARPOL and the date they entered into force internationally [246].

| Annex | Annex Title | Entry into Force |
|:---:|:---:|:---:|
| I | Regulations for the Prevention of Pollution by Oil | 2 October 1983 |
| II | Regulations for the Prevention of Pollution by Noxious Liquid Substances in Bulk | 2 October 1983 |
| III | Regulations for the Prevention of Pollution by Harmful Substances Carried by Sea in Packaged Form | 1 July 1992 |
| IV | Regulations for the Prevention of Pollution by Sewage from Ships | 27 September 2003 |
| V | Regulations for the Prevention of Pollution by Garbage from Ships | 31 December 1988 |
| VI | Regulations for the Prevention of Air Pollution from Ships | 19 May 2005 |

Currently the IMO includes 174 member states and three associate members [247], however, not all members are signatory to MARPOL, nor to all the annexes (Table 3).

**Table 3.** The number of IMO member states signatory to each of the six MARPOL annexes and the proportion of the world's merchant shipping gross tonnage this represents [248].

| Annex | Number of Member States | % World Tonnage [1] |
|:---:|:---:|:---:|
| I and II | 159 | 98.95 |
| III | 149 | 98.40 |
| IV | 145 | 96.33 |
| V | 154 | 98.56 |
| VI | 98 | 96.76 |

[1] Percent of the gross tonnage of the world's merchant shipping.

While MARPOL is the principal ship-sourced pollution regulation, the IMO also has several other conventions relating to marine pollution and other matters of relevance to managing boating and shipping operations [249]:

- Convention on the International Regulations for Preventing Collisions at Sea (COLREG), 1972
- International Convention on Maritime Search and Rescue (SAR), 1979
- International Convention for Safe Containers (CSC), 1972
- International Convention Relating to Intervention on the High Seas in Cases of Oil Pollution Casualties (INTERVENTION), 1969

- Convention on the Prevention of Marine Pollution by Dumping of Wastes and Other Matter (LC), 1972 (and the 1996 London Protocol)
- International Convention on Oil Pollution Preparedness, Response and Co-operation (OPRC), 1990
- Protocol on Preparedness, Response and Co-operation to pollution Incidents by Hazardous and Noxious Substances, 2000 (OPRC-HNS Protocol)
- International Convention on the Control of Harmful Anti-fouling Systems on Ships (AFS), 2001
- International Convention for the Control and Management of Ships' Ballast Water and Sediments, 2004
- The Hong Kong International Convention for the Safe and Environmentally Sound Recycling of Ships, 2009.

The IMO's overall mission is to promote safe, secure, environmentally sound, efficient, and sustainable shipping through cooperation. The organization aims to achieve its mission by implementing the highest practicable standards of maritime safety and security, efficiency of navigation, and the prevention and control of ship-sourced pollution. Additionally, as part of this overall process, consideration is given to the associated legal matters and the effective application of the various IMO instruments, with the ultimate goal of universal and uniform implementation [250]. Additional IMO works toward environmentally sound shipping operations and the protection of the marine environment is through the designation of Particularly Sensitive Sea Areas (PSSAs). A PSSA is an area that requires special protection because of its significance for recognized ecological, socio-economic, or scientific reasons, and which may be vulnerable to damage by international maritime activities [251]. The IMO has guidelines for designating PSSAs that include measures to allow areas to be designated if they fulfil a number of criteria, including ecological criteria (e.g., rare or unique ecosystem, diversity of the ecosystem or vulnerability to degradation either by natural events or human activities), social, cultural, and economic criteria (e.g., significance of the area for recreation or tourism), and scientific and educational criteria (e.g., research or historical value) [251]. Areas approved as PSSA, are designated specific measures used to control the maritime activities in that area (e.g., routing measures, strict application of MARPOL discharge and equipment requirements and installation of Vessel Traffic Services) [251]. Currently, there are 15 designated PSSAs worldwide [252], e.g., Great Barrier Reef (1990), Florida Keys (2002), Wadden Sea (2002), Canary Islands (2005), Galapagos Archipelago (2005), Baltic Sea area (2005), Papahānaumokuākea Marine National Monument (2007), Strait of Bonifacio (2011), Tabbataha Reefs National Park (2017).

Because of the vast number of different jurisdictions globally (from international through to national, regional, and local levels), describing in detail all of the various legislative instruments and other measures used in each jurisdiction to manage boating- and shipping-related impacts would be a monumental and time-consuming task that would presumably be outdated by the time of completion. As such, Australian, and cases from its states and territories, are presented as example jurisdictions used to describe various legislative instruments and measures used to manage boating- and shipping-associated environmental impacts. Such an approach has been used by others, e.g., see Byrnes et al. [16], for justification.

Background to the Australian System of Government

In 1901, six British colonies united to form the country of Australia. The colonies became Australian states and a federal Parliament was created with the power to make laws regarding national matters. The Australian Constitution, the set of rules by which Australia is run, establishes how the federal and state parliaments share the power to make laws [253]. The national or central government of Australia is generally referred to as the Federal Government, Commonwealth Government, or Australian Government [254]. However, federal Parliament alone does not make all laws across the nation. The following three levels of government work together to provide various laws and services required:

1. Federal, Commonwealth, or Australian -Parliament creates laws for the nation;

2.   Six state and two mainland territory parliaments create laws for their respective state or territory;
3.   Over 500 local councils (shires) across Australia create local laws (by-laws) for their region or district [253,255].

Australia is one of 97 Member States that is signatory to all six MARPOL annexes [248]. This is achieved at the national level principally through the *Protection of the Sea (Prevention of Pollution from Ships) Act 1983* (PSPPSA) and at the various state and territory levels through [256]:

- *Marine Pollution Act 2012* (New South Wales);
- *Protection of Marine Waters (Prevention of Pollution from Ships) Act 1987* (South Australia);
- *Pollution of Waters by Oil and Noxious Substances Act 1987* (Western Australia);
- *Pollution of Waters by Oil and Noxious Substances Act 1987* (Tasmania);
- *Pollution of Waters by Oil and Noxious Substances Act 1986* (Victoria);
- *Transport Operations (Marine Pollution) Act 1995* (Queensland);
- *Marine Pollution Act 1999* (Northern Territory).

While Australia has implemented all annexes of MARPOL at the national level, not all Australian states and territories have adopted all annexes into their own state requirements (Table 4), particularly Annex VI which is yet to be adopted by any Australian state or territory [256].

**Table 4.** Annexes of MARPOL adopted by each state and territory of Australia [256].

| State | Annex I Oil | Annex II Chemicals | Annex III Packaged | Annex IV Sewage | Annex V Garbage | Annex VI Air Pollution |
|---|---|---|---|---|---|---|
| QLD | √ | √ | √ | √ | √ | X |
| NSW | √ | √ | √ | √ | √ | X |
| VIC * | √ | X | X | X | √ | X |
| TAS | √ | √ | √ | √ | √ | X |
| SA | √ | √ | √ | X | √ | X |
| WA | √ | √ | X | X | X | X |
| NT | √ | √ | √ | X | √ | X |

QLD = Queensland; NSW = New South Wales; VIC = Victoria; TAS = Tasmania; SA = South Australia; WA = Western Australia; NT = Northern Territory; * = Operational aspects only.

Readers are referred to the Australian Maritime Safety Authority (AMSA) [257] for an overview of the various discharge requirements for boating and shipping operations in Australia. Theoretically the MARPOL regulations include most the pollution-related aspects of boating- and shipping-associated environmental impacts. However, most of the marine pollution control provisions of MARPOL, and legislated by various individual jurisdictions globally, are aimed at large-sized vessels exceeding 400 gross tons or vessels surveyed for >15 people, and for some provisions, limited only to vessels that are engaged in an international journey (i.e., predominantly the "shipping" aspects of boating and shipping impacts). Thus, it can sometimes be difficult to determine exactly if, and to what extent, such provisions apply to other vessels. There may be situations whereby subtle yet important differences in the phrasing and language used in legislation may lead to complex situations and confusion, for any of the various boating and shipping operator groups. For example, in Australia, the *Protection of the Sea (Prevention of Pollution from Ships) Act 1983* includes no specific definition for "ship" with the exception of general references to terms included under MARPOL, while under parts of Queensland's respective legislation, the *Transport Operations (Marine Pollution) Act 1995* (TOMPA) (e.g., sections 49 and 50), "declared ships" are specified (i.e., ships that have a fixed toilet and are either a class 1B, 1C, 1D, 1E, 4C, 4D, or 4E domestic commercial vessel, or a Queensland regulated ship designed to carry >12 passengers). These "declared ships" are subjected to higher sewage discharge standards under sections 33–39 of the *Transport Operations (Marine Pollution) Regulation 2018* (QLD). This legislation also offers comprehensive details regarding sewage treatment levels and associated discharge areas and other relevant information. There are also special rules and requirements applied

to specific areas within Australia, such as marine parks in Queensland under the *Marine Parks Act 2004* (QLD) and accompanying 2017 regulation or those in Commonwealth waters such as the GBRMP under the *Great Barrier Reef Marine Park Act 1975* and its associated regulations and zoning plans.

To aid in the interpretation of the various legislative requirements in Australia, and avoid potential confusion, under the *Commonwealth of Australia Constitution Act* (The Constitution), when a law of a State is inconsistent with a law of the Commonwealth, the latter shall prevail, and the former shall, to the extent of the inconsistency, be invalid (section 109 *Commonwealth of Australia Constitution Act*)—although such a scenario would likely require a determination from the High Court of Australia. As such, if state-based legislation stipulating management measures equivalent to (or more stringent than) requirements under the PSPPSA exist, then the state-based legislation should prevail.

However, jurisdiction of the State is limited to coastal waters, designated as typically three nautical miles from the low water mark. Depending upon the geographical location, type of activity and season, some boating and shipping operators, such as those engaged in recreational fishing or diving activities, may extend their operations well beyond coastal water limits. In some situations, operators may travel across water bodies covered by various management authorities that would require consultation of various discharge provisions of marine pollution and conservation legislation at both the federal and state jurisdictional level.

Various zoning plans can also contain provisions regarding certain boating and shipping activities, such as vessel anchoring practices, wildlife disturbances, and operational restrictions. For example, in Queensland, the *Marine Parks (Moreton Bay) Zoning Plan 2019* regulates certain activities within the Moreton Bay Marine Park (MBMP) under a four-tiered zoning plan of (i) General Use, (ii) Habitat Protection, (iii) Conservation Park, and (iv) Marine National Park zones, with each zone providing a greater level of protection than the preceding zone. Such zoning plans state permissible activities within each zone. Further, if such zoning plans do not make mention as to whether or not certain activities are permitted (with or without necessary the permissions), then those activities are generally deemed to be prohibited (e.g., using a vessel as a permanent dwelling within the marine park).

Additional matters considered under the example zoning plan include restrictions placed on commercial endeavors and designated areas within the marine park (see *Marine Parks Regulation 2017* (QLD)). Designated areas include "go slow" (speed reduction), no-anchoring, and mooring areas. Go slow areas prohibit the operation of a vessel in either a planning or non-displacement mode or to engage in motorized water sports. Additionally, in designated turtle and dugong go slow areas, the operation of a vessel in a way (or at a speed) that may result in any animal strikes, or for vessels >8 m in length to exceed a speed of 10 knots, is prohibited. Areas designated for mooring, permit mooring activities to be undertaken but with certain restrictions including mooring design, installation and use (e.g., an EFM). Areas prohibiting anchoring aim to protect benthic habitats by making it an offence to anchor in such areas. Under the current zoning plan there are 17 go slow, 45 mooring, and 3 no-anchoring areas within the MBMP.

Although limited in its application to certain waters of Queensland's Gold and Sunshine Coasts, the *Transport Infrastructure (Waterways Management) Regulation 2012* (QLD) provides several measures to manage potential boating- and shipping-related environmental impacts. For example, prohibitions on anchoring within 30 m of an approved structure or a vessel moored to an approved structure, and waters where vessels cannot be anchored or moored for (a) more than 24 consecutive hours in any 30 day period, (b) more than 7 consecutive days in any 60 day period, and (c) within 1 nautical mile of the 1 place for more than 7 consecutive days in any 60 day period (refer Part 3). There is also some specific living on board requirements. For example, in Gold Coast waters, the owner or operator of a vessels must not live, or allow anyone else to live, on board, at, or within 3 nautical miles of, the 1 place for more than 7 consecutive days in any 60 day period (unless the vessel is at a marina with toilets and washing facilities).

Whereas at the Sunshine Coast, vessel owners or operators must not live, or allow another person to live, on board, whether temporarily, intermittently, or permanently, without a living on board

approval. Additionally, they must also ensure that the vessel has a waste holding system, regardless of whether it is occupied or not, and that none of the contents are discharged into the water. Furthermore, vessel operators must ensure that a fixed or mobile pump-out facility is used to empty the contents of the waste holding system and keep written records of all discharges, specifically (a) the date when, and the place where, the contents of the system were discharged, and (b) the quantity of the contents discharged. Reference to a waste holding system in this context is taken to be a waste holding tank connected to each source of sewerage or wastewater on the vessel.

Queensland's *Transport Operations (Marine Safety) Regulation 2016* also contains a provision that appears to be tailored toward the management of potential boating and shipping impacts whereby a person must not operate a ship at a speed at which the ship's wash is reasonably capable of causing a marine incident or damage to the shoreline (refer section 83 *Transport Operations (Marine Safety) Regulation 2016*). Perhaps this may be viewed as a somewhat rudimentary attempt to manage environmental (and human health) impacts as it leaves it up to the operator to use "common sense" to determine (a) whether their vessel's wash is likely to be causing a marine incident or shoreline damage, and (b) what speed they should be operating at to ensure that such an outcome does not occur (see Section 3.4 for discussion on "common sense").

Additional government regulations controlling aspects of pollution from vessels can also be captured within the general pollution control legislation, such as the regulation of TBT-based (antifouling) paints sale or use under provisions of the *Environmental Protection Act 1986* and *Environmental Protection Regulations 1987* in Western Australia. In Queensland, similar provisions were initially contained in both the *Chemical Usage (Agricultural and Veterinary) Control Act 1988* (QLD) and *Chemical Usage (Agricultural and Veterinary) Control Regulation 1989* (QLD). However, the regulation was replaced in August 1999 by legislation that no longer made reference to TBT. A convention banning the use of TBT on all vessel sizes was later passed by the IMO (October 2001). As per other international conventions passed by the IMO, entry into force requirements are met a year after obtaining the required number of Parties for ratification with a "phase-in" progression. The IMO resolved that by 1 January 2003, all parties should implement a ban on TBT use within their own jurisdictions. Within Australia, the Australian Pesticides and Veterinary Medicines Authority (formerly the National Registration Authority for Agricultural and Veterinary Products) effectively deregistered all products comprising TBT from 31st March 2003, with its possession, supply or use beyond the 31st of July 2003 banned. Conditions for entry into force were met with the convention entering into force within Australia and internationally on 17th September 2008. This was enacted through the International Convention on the Control of Harmful Anti-fouling Systems on Ships 2001 (AFS Convention) and the *Protection of the Sea (Harmful Anti fouling Systems) Act 2006* (Cth), respectively.

Most of the abovementioned pieces of legislation are typical examples of direct regulatory instruments, in that sanctions for non-compliance are imposed (e.g., fines) while ongoing policing is required to deter vessel operators from ignoring the management measures aimed at preventing environmental impacts. These instruments capture various aspects of boating- and shipping-related pollution and disturbances. However, substantial overlap and variations in the detail of these instruments can be a source of confusion for recreational and commercial vessel operators undertaking activities in Australian waters under the jurisdiction of, and potentially even beyond, both State and Commonwealth authorities. This may become even more problematic for those operators that use their vessels across state borders where operators must comply with multi-jurisdictional State(s) and Commonwealth legislative requirements.

### 3.3. Supplementary Management Measures

Direct regulations have been adopted for a variety of environmental impacts associated with boating and shipping operations, either industrially, commercially, or recreationally based. The success of these mechanisms hinges on the level to which compliance can be efficiently and effectively monitored and enforced. Costs and difficulties associated with such monitoring and enforcement may be justified

and, to an extent, recouped (e.g., the various levies and fees imposed by the *Protection of the Sea (Shipping Levy) Act 1981* (Cth)) through the surveillance of larger (commercial) vessels (i.e., again the shipping part of the boating and shipping categories). Most accidents or incidents resulting from negligence and/or breaches of pollution control provisions by this larger vessel group, has the very real potential to cause substantial environmental harm. However, from a technical standpoint, these larger vessels can be more effectively monitored, as they are typically restricted to defined commercial shipping routes, maintain relevant pollutant register often times require a pilot.

In contrast, it is much more difficult to monitor smaller vessels as they are not generally bound to defined shipping routes, nor are they required to maintain pollutant registers as they would tend to be <400 gross tons. While the per vessel potential environmental impacts resulting from smaller vessels is generally much lower, their overall potential for cumulative impacts is considerable. Unfortunately, the costs associated with the broadscale monitoring and enforcement of compliance with government regulations for smaller vessel operators is difficult for many regulatory authorities to justify [240]. For example, such as the physical difficulties, time, expense, and resources required to monitor compliance with the living on board requirements of the *Transport Infrastructure (Waterways Management) Regulation 2012* (QLD) in more remote locations, particularly when attempting to establish that breaches of the regulation have occurred.

Governments are becoming increasingly reliant on industry-based management instruments to support and enhance their existing regulatory frameworks. Gunningham et al. [241] highlights that the type and combination of adopted instruments is dependent on several factors including the existing regulatory framework, the environmental problem or issue to be addressed, the industry structure and the various roles of industry and key stakeholders and interest groups.

### 3.3.1. Industry-Based Measures

The following section aims to explain and describe various aspects of industry-based environmental management instruments for boating and shipping operations by way of a "quasi" case study example of the Australian tour boat industry based on works by Byrnes et al. [16], Byrnes, [195] and Byrnes and Warnken [258]. The Australian commercial tour boat industry is a diverse and geographically diffuse sector with around 1500 individual members with a combined fleet of approximately 3800 vessels, some professional and fully commercial tourism businesses, while others are scarcely distinct from the private recreational boating sector [16]. Environmental impacts arising from tour boat operations are generally difficult to control through existing legislation and regulation alone. Policing the relevant provisions at a large scale usually becomes too costly to justify their expense in relation to their likely magnitude of impact. As an alternative to legislation and regulations imposed by government and management authorities, industry self-regulation has been promoted to minimize tour boat-related impacts [258].

Moral suasion measures such as self-regulation is frequently promoted as an addition to institutional regulation by government authorities [259,260]. Self-regulation is necessary for industry sectors not adequately covered by legislation, which are therefore ultimately responsible for regulating their own impact on the environment. Since the marine environment is the principle asset for boat operators, self-regulation is clearly in their interest [258]. Formal adoption of environmental management guidelines is a form of industry self-regulation. This is generally in the format of a set of technical/operational guidelines that outline the most effective ways of avoiding or mitigating adverse effects of human activities on the natural environment. For the various sectors of the boating community these guidelines could be introduced through one or several industry associations, which may then adopt the guidelines and encourage their members to implement them [258].

### 3.3.2. The Australian Tour Boat Industry: Case Example

The successful implementation of industry-based environmental management measures by Australian tour boat operators appears somewhat limited for various reasons. First, the majority

of businesses operating in the industry tend to be small or very small in size with ≤10 full-time employees [261]. Like many small businesses, they often are economically marginal with restricted staff and economic resources at their disposal, making the broadscale application of elaborate environmental management systems unlikely (see Delmas [238] and del Brio and Junquera [262]). On a similar economic rationale, nation-wide education and training programs would likely only be successful if they are developed, established, and supported by government agencies or relevant key industry players with the necessary resources and willingness to commit to the successful implementation of any such program. Similarly, the cost argument seemingly also applies when discussing various other initiatives, programs, or schemes such as green taxes, corporate environmental reporting, pollution inventories, and performance bonds. It is also unlikely that instruments including product certification schemes or tradeable pollution rights would apply to smaller vessels because of the inherent difficulties in quantifying their individual and overall pollutant emissions or physical disturbances to habitats and wildlife.

The development of various green eco-labels for tour operators have taken place since the mid-1980s [263], including those for marine and vessel operators (e.g., eco-tourism accredited operator: Ecotourism Australia's Eco Certification scheme [264], or Green Globe's certification [265]). However, industry penetration levels were minimal [266]. This poor uptake level may have resulted from the many different and competing certification schemes based on vastly different standards that lacked a functioning principal accreditation body for validation.

Historically, threats of major civil liabilities from pollution incidents/accident or disturbances arising from tour boat operations have been too small to create enough interest or momentum for operators to form industry-based control and insurance systems. Large cruise ships being the exception, tourism-based vessels are characteristically at the smaller end of the size spectrum (refer to the vessel characterization in Byrnes et al. [16] for details), thus any environmental impact resulting from an individual tourism vessel is likely to be relatively smaller in comparison to larger ships (sewage loads maybe being the exception). It is unlikely that these typically smaller scale impacts would affect the economic prosperity or livelihoods of sectors of the community large enough to warrant the initiation of class actions or government instigated actions seeking damages (see Fisher [267]). Therefore, it appears that the sole threat of civil liability does not promote widespread self-regulation aimed at increasing the tour boat industry's environmental performance levels.

A typically common type of industry self-regulation is the development of environmental based standards, codes of practice or guidelines implemented by industry members or sectors, either voluntarily or resulting from intervention by government [241]. While government bodies usually provide the (general) goals, rules, or requirements for such schemes, they do provide scope for industry participation into the overall process, including potential enforcement mechanisms (if any). Theoretically, this approach should lead to the establishment of more practical standards, higher levels of compliance, and effective enforcement given that the setting of standards and monitoring of compliance are practitioner-based, with greater knowledge and experience of their own industry's operations (e.g., size, structure, constraints). Similar to EMSs, a primary objectives of these environmental management guidelines (or codes of practice) is to improve the widespread levels of operator compliance with existing regulatory requirements. Additionally, such guidelines have the ability to provide innovative and practical solutions to environmental impact management that should promote an enhanced level of performance well beyond that imposed by governmental regulators.

The tour boat industry has a slightly different setting to other small to medium enterprises, whereby there are existing regulations for larger vessels, which, if applied to relevant tour boat vessels/operations, have the potential to minimize or prevent the majority of their potential impacts. A case could also be put forward that the shipping industry would have provided some input into the development of the various MARPOL conventions through lobbying and representations to the IMO's Marine Environment Protection Committee. Thus, the existing international regulatory regime provides environmental management standards founded on a degree of industry participation.

### 3.3.3. Policy Instruments

Managers also need to be cognizant of the various policy instruments available and the role that each of them may play in helping to achieve a positive environmental management outcome.

There are five common types of instrument described by Althaus et al. [268] for the delivery of the desired policy objectives: (i) policy through advocacy, (ii) network, (iii) money, (iv) government action and (v) law. Using information available to governments, policy through advocacy uses education or persuasion to achieve desired policy goals, outcomes or objectives. For boating and shipping operations this may include education regarding specific environmental impacts, causes and consequences, and available management measures. Cultivating and leveraging relationships both within and across government with external partnerships (e.g., in a boating and shipping context this may include industry organizations) to achieve desired policy goals, objectives, and outcomes is an example of policy through networks. Spending and taxing powers used to influence activity beyond government to achieve desired policy outcomes are typically a policy through money approach. Practical examples include financial incentives offered to marina or harbor managers to encourage the installation and management of appropriate waste reception facilities for ship-sourced pollutants, such as skip bins, waste oil tanks for bilge water residues or discarded engine oil, or pump-out facilities for sewage. Policy through direct government action represents service delivery through public agencies, to achieve the desired policy goal, objective, or outcome. For example, ensuring that the appropriate implementation of onshore waste reception facilities for ship-sourced pollutants (e.g., installed, operated and maintained). Finally, policy through law encompasses the use of legislation, regulation, and official authority, in order to achieve desired policy outcomes. As previously presented, many laws already exist for the management of boating- and shipping-related environmental impacts (e.g., MARPOL, TOMPA, PSPPSA, *Marine Parks Act 2004* (QLD)).

Managers need to approach specific environmental issues in a clear, logical, and strategic way. Key to this is being able to clearly identify the specific environmental management issue to be addressed and then consider all types and ranges of potential instruments available. This includes understanding the full range and scope of regulatory alternatives and their respective pros and cons, such as that provided by Queensland Treasury [269].

A good example of the type of comprehensive approach needed to address complex environmental issues is the principle-based approach used by Australian governments (Principles of Best Practice Regulation) [270]:

- Before addressing the problem, establish a clear case for action;
- Consider a range of policy options, including self-regulatory, co-regulatory and non-regulatory methods, whilst assessing each of their individual costs and benefits;
- Implement the option that creates the greatest net benefit for the community;
- In agreement with the Competition Principles Agreement, competition should not be restricted by legislation unless it can be established that (a) benefits of the restrictions to the community, as a whole, outweigh the costs; (b) aims of the regulation can be achieved only by restricting competition;
- Deliver effective guidance to appropriate regulators and regulated groups in order to ensure that the intentions of the policy and the expected compliance requirements of the regulation are clear;
- Ensure that regulation continues to remain relevant and effective;
- At all stages of the regulatory cycle provide, effective consultation with affected key stakeholders;
- Government action should be effective and proportional to the issue being addressed.

### 3.4. "Common Sense" Management Measures

There is scope for the implementation of (simple) self-imposed measures with the aim of minimizing the potential for environment impacts related to either boating or shipping operations, regardless of whether or not there is any involvement or overarching control by government or industry.

In the context of this review, such self-imposed measures are colloquially referred to as "common sense," though, caution should be applied regarding the use of such terminology.

In Australia, largely as a result of national anti-littering campaigns (e.g., the "Keep Australia Beautiful" campaign [271]—ongoing anti-littering messaging initiated in the 1970s), it would be difficult for community individuals to not understand that it is neither appropriate, nor "common sense," to dispose of garbage irresponsibly (i.e., it should only be disposed into a designated garbage or recycling bin). As such, it is a reasonable expectation that vessel operators (e.g., commercial, recreational or tourism based) also exhibit appropriate management behaviors regarding their vessel-sourced garbage and debris.

Examples of Practical and Behavioral Measures to Manage Environmental Impacts

There are simple self-imposed practical and behavioral measures that boating and shipping operators can implement to minimize environmental impacts in relation to the way they operate vessels. Examples are provided, however it should be noted that this information is not an exhaustive compilation, rather it presents some examples of "common sense" options that can be implemented by boating and shipping operators with relatively little effort and/or expense to reduce environmental impacts. The various examples are presented under the headings of "Practical Measures and Behavioral Measures".

• Sewage

Measures implemented for the appropriate management of sewage largely relate to where, when, and how discharges occur and the on board sewage management equipment (e.g., toilet with treatment system and/or a holding tank, use of onshore discharge, no toilet, etc.).

Practical Measures

One of the simplest practical sewage management measures is the option of having no toilet on board a vessel, which would "theoretically" equate to no discharge. This is because discharges of sewage, especially feces, is less likely simply due to practical constraints (i.e., an individual would be required to urinate or defecate off the vessel or in a collection container to be discharged into the receiving environment). Similarly, having a toilet connected to an on board sewage holding tank that is discharged ashore is the soundest environmental option (i.e., from a combined operator/environment viewpoint) as the sewage is contained and discharged ashore before reaching a municipal sewage/wastewater treatment plant. This approach reduces the likelihood of sewage discharges from a vessel, while also permitting a more all-inclusive approach to the management of sewage for any given regional sewage management strategy.

An on board toilet and associated treatment system is, in terms of limiting environmental impacts, theoretically better than discharging untreated sewage directly into the receiving environment. However, it should be noted that present-day on board treatment systems tend to afford only a rudimentary level of sewage treatment in comparison with large-scale terrestrial sewage treatment plants, particularly smaller units (i.e., sewage is macerated, chemically treated and directly discharged into the receiving environment–where regulations permits) (e.g., [272]) with treatment levels principally focused on human health (e.g., the removal of pathogens) and not the environment (e.g., the removal of nutrients). Terrestrial sewage treatment plants are also staffed by a team of professionals (including those with a scientific background) that regularly and continuously monitor the state of the plant and the quality of the wastewater prior to discharge, whereas an on board sewage treatment system fitted to a vessel operating in Queensland need only have the levels of sewage quality characteristics assessed at least annually for the first two years (declared ship) or at least once in the first five years (another

ship) and then at least every two years thereafter (for both) [1]. However, when properly operated and maintained, an on board sewage treatment system affords a better alternative than discharging raw sewage, while also demonstrating more positive operator behavior.

A toilet/holding tank combination which discharges overboard does provide vessel operators with a level of discretion as to where (when) the untreated sewage is discharged. Operators may elect to hold the sewage on board when in areas discharges present a risk to the environment and/or human health (e.g., coral reefs, oyster leases, and recreational contact areas, etc.,) and discharge once their vessel has travelled an appropriate distance away from such areas. In regions lacking legislative requirements for sewage discharges, a decision regarding the appropriate distance from sensitive areas is reliant upon the understanding of the vessel operator with regard to the environmental (and human health) issues involved (i.e., "common sense").

A toilet discharging directly overboard represents the simplest on board sewage management option (including a toilet) for operators of boats or ships. However, this scenario also presents the greatest potential for environmental harm because: (i) Sewage can be discretely discharged (i.e., from a closed door facility with discharge outlet almost exclusively below waterline); and (ii) the operator has minimal control regarding where and when sewage loads are discharged, with the exception of (a) preventing access to the toilet facility or (b) by actively dictating to passengers amenity usage periods.

Behavioral Measures

Behavioral measures that may be employed for appropriate sewage management with regard to when, where, and how discharges take place include avoiding discharges in areas deemed environmentally sensitive, shallow, characterized by low energy, poorly flushed (e.g., marinas, anchorages, etc.,), or areas where recreational contact within the waterbody is likely to occur, regardless of any specific regulations. If a vessel is not fitted with a holding tank(s), operators may request, or direct passengers to cease using the facility while in such sensitive areas. Prior to any vessel trip, operators may also actively encourage pre-departure use of terrestrial-based toilet facilities to reduce the incidences of toilet use during any given trip.

● Garbage/Debris

The predominant sources for garbage in receiving aquatic environments sourced from boating- and shipping-associated activities include items blown (accidental) or thrown overboard (deliberately, e.g., cigarette butts, food scraps, etc.,) during trips. Additionally, debris in receiving aquatic environments commonly are the result of lost vessel parts and/or equipment.

Practical Measures

Practical measures that may be adopted to reduce the potential impacts from garbage include decisions such as to ensure access to a suitable number of easily identifiable garbage bins and/or ash trays on board, that these are clearly labelled and that no garbage is deliberately discarded overboard. Additionally, to decrease the potential impacts of garbage, vessel operators should also ensure that all on board garbage is collected and appropriately stored for later onshore disposal.

Behavioral Measures

Operator behavior has an important role to play in minimizing garbage/debris-related environmental impacts associated with boating and shipping operations. This may include any, all, or a combination of the following example measures. Reminding passengers not to discard garbage items overboard. Where possible minimize packaging to reduce the potential of garbage being blown/thrown overboard. In the event of crewed vessels, it is essential to ensure all crew members are appropriately informed on (and understand) the importance of providing vessel passengers

---

[1] See *Transport Operations (Marine Pollution) Regulation 2018* (QLD), section 43 "Maintenance and assessment of treatment systems for Ships."

with examples of appropriate garbage management practices thus avoiding copy-cat behavior of inappropriate behavior.

Regarding garbage-related impacts from the vessel itself, boating and shipping operators should ensure regular inspection and maintenance regimes for the entire vessel including on board facilities and equipment to minimize the potential for vessel parts being dislodged or broken off during severe weather events. Additionally, vessels should be operated in an appropriate manner (i.e., suitable speeds, changes of direction, course plotting, navigation, etc.,) to minimize the likelihood of incidents or accidents that may result in the loss of vessel parts and/or equipment.

● Hydrocarbons

Apart from major incidents and accidents, the major pathways for hydrocarbons into aquatic environment from boating- and shipping-related activities includes leaks and/or spills either by: (i) Finding their way into the bilges and other collection spaces on board vessels, which are then discharged overboard as a hydrocarbon water mixture via on board pumps; or (ii) discharged directly overboard into the receiving waters. There are several measures that boating and shipping operators can implement to help avoid such pathways for hydrocarbons entering aquatic environments.

Practical Measures

Practical measures to minimize the potential for hydrocarbon-based environmental impacts include the use of drip trays ("catch-alls") under engines or other of hydrocarbon sources or the use of hydrophobic absorbent materials (i.e., sorbent pads, mats, socks or sausages) in the bilges or other spaces where hydrocarbons or hydrocarbon-water mixtures are likely to collect. While it may be part of a conscientious and diligent operator's general routine, there are also some simple "common sense" practices such as ensuring that engines are well monitored and maintained and that caution and vigilance is exercised during re-fueling procedures (i.e., to ensure that spills, overflows, etc., do not occur, and if so, quick response measures are initiated). Further measures may include changes to utilize engine technology from two-stroke outboard engines to four-stroke, or modern two-stroke outboards with enhanced emission reduction technology. All on board waste hydrocarbons should be collected and stored on board in appropriate storage vessels for subsequent disposal at designated onshore waste collection points.

Behavioral Measures

Operator behavior aimed at preventing or minimizing potential hydrocarbon-related impacts include ensuring that all vessel staff are well educated on the importance of correct and safe procedure and remaining vigilant at all times during re-fueling and also that they have a sound understanding of necessary actions in the event of a hydrocarbon spill. As is typical with almost all of the operator behavioral measures that can be implemented to minimize potential environmental impacts, it is essential that vessels are operated in an appropriate manner (in regard to speed, direction, navigation, keeping a proper look out, etc.,) to minimize the potential for the types of accidents/incidents that may result in hydrocarbon losses into the aquatic environment.

● Greenhouse Gas Emissions

The primary origin of vessel-sourced GHG emissions is the burning of fossil fuels associated with internal combustion engines that are used for propulsion through the water or to generate electricity for on board electrical plant, equipment, and appliances.

Practical Measures

To reduce GHG emissions in a practical manner typically relates to lowering the total amount of fossil fuels burnt while still achieving the required operational outcome. Avenues to achieve such outcomes include individual engine efficiency in addition to the overall efficiency of hull design/engine specification combination (i.e., improved the overall engine/vessel efficiency = reduced GHG emissions). Excessively over-powered engine/s with associated high fuel consumption rates, fitted to vessel whose function is to carry low per trip passenger numbers, is a situation that will result in unduly high GHG

emissions and should be avoided. Further, upgrading (older) less efficient engines with more modern technologies that have better fuel efficiencies is an additional practical GHG emissions reduction measure. A side-benefit of this practice is that the more modern and efficient engines are generally physically smaller and lighter, providing additional savings on space and weight (to further reduce fuel use and subsequent GHG emissions). Simply adhering to regular engine servicing and maintenance schedules should also ensure that engines continue to operate at optimal efficiency.

Behavioral Measures

Behavioral measures that can be implemented to reduce excessive fuel consumption include operating vessels in the most fuel-efficient manner with regard to operational aspects such as speed, acceleration, and changes of direction. This is an important operator behavioral trait to be recognized, as is having systems and processes in place to ensure that engine servicing and maintenance schedules are adhered to, as ways to help reduce excessive GHG emissions.

• Antifouling Agents

There are alternatives available to impede the rate at which fouling organisms colonize submerged vessel surfaces that are less harmful to the environment (and non-target organisms) than those traditionally adopted. Such alternatives include the use of non-toxic, non-stick, fouling-release coatings [273], fouling-release fluorinated polymers, and silicon and amphiphilic surfaces [274], biodegradable polymers, and natural antifoulants such as butenolide derived from marine bacteria [275] and various naturally based biodegradable polymers, such as from cashew nut shell liquid, incorporating natural antifouling products, for example zosteric acid (sourced from marine plants), isocyanoterpenes (sourced from sponges and nudibranchs) and mangrove tannins (sourced from the mangrove plants) [276].

Practical Measures

Practical measures that operators may practice primarily include not applying known harmful antifoulants to the hulls and other submerged parts of their vessel (e.g., rudders, outboard motor keels, sterndrive engine drive legs and the hydrojets of jet propulsion systems). Alternative practical measures include regular removal of fouling organisms by manual hull scrubbing, the use of an in-water skirt filled with freshwater when moored (for those operating in seawater) or removing vessels from the water when they are not being used (e.g., at the conclusion of activities for each day or trip) and storing them in marina dry stacks, dry dock moorings that raise the vessel out of the water once moored (e.g., [277]) or on boat trailers, for example. Obviously, many of these practical measures are restricted to the smaller end of the boating and shipping size spectrum.

Behavioral Measures

Operators or vessel contractors may investigate various antifouling agents offered and use the agent which best suits their needs and circumstances, while minimizing impacts on non-target organisms while providing the optimal operational outcome at a relatively similar financial burden. Apart from the scientific literature outlined above, a "common sense" approach could include an online search to find useful and relevant information regarding environmentally friendly antifouling options (e.g., [278]), that may provide an ideal starting point.

• Physical Disturbance of Habitats and Fauna

Physical disturbances to habitats and fauna are difficult to appropriately manage with minimal specific regulation applicable in most scenarios (outside of the previously mentioned "go slow" and "no anchoring" zones of marine parks). It is a requirement for commercial vessel operators to undergo a comprehensive formal training and assessment process to gain a commercial operator's license. As such, this should ensure a relatively competent understanding of appropriate vessel operations that should subsequently minimize vessel groundings. Some marine park managers may manage the impacts of anchoring through the implementation of specific "no anchor" zones. Additionally, the installation of moorings (especially if they are EFMs–see below) may also reduce the impacts of

vessel anchoring practices. Furthermore, marine parks may also designate specific "go slow" zones to reduce the potential for vessels striking fauna. Unfortunately, such measures are not generally available to managers outside the boundaries of specific marine protected areas.

Regarding EFMs for example, since 2009, Healthy Land and Water (HLW) in southeast Queensland, has been trialing the effectiveness of various EFM designs in designated mooring areas of the MBMP [279]. Since 2012, they have been collaborating with State and Federal governments to design and roll out an EFM replacement program in areas with priority seagrass meadows to greatly reduce the impact caused by traditional block and tackle moorings on the marine environment (i.e., anchor chain damage). More than 230 traditional block and tackle moorings have been replaced with EFMs to date [279]. Additionally, more recently (29th May 2020), HLW partnered with the Gold Coast Waterways Authority (GCWA) to replace 22 traditional block and tackle moorings with EFMs in the Jacobs Well and Cabbage Tree Point designated mooring areas of the MBMP [280]. The installation of EFMs is supported by GCWA's "Buoy Mooring Management Strategy" which identifies managing the environmental impacts of buoy moorings as a priority [281,282].

Practical Measures

There are practical measures that vessel operators may implement to reduce the potential anchoring-related impacts, including any, all, or a combination of any of the following. The use of established moorings or appropriate amenities/structures, when and where available rather than anchoring. Proactive vessel positioning during times of anchoring to ensure the lowest impact to substrate or benthic habitat (i.e., sand rather than coral or seagrass). Additionally, vessel operators may actively seek involvement with any existing EFM initiative. Furthermore, operators may fit suitable sound reducing devices to limit vessel-sourced noise, while also obeying speed restrictions to reduce negative impact on the surrounding fauna.

Behavioral Measures

Behavioral measures are a key factor in the reduction of physical disturbance of habitats and fauna from boating and vessel operations. Important behaviors include vessel operation in sufficiently deep waters to avoid grounding and limit impacts to soft-bottoms corals and seagrass meadows, for example. Additionally, vessel operation should be undertaken at suitable speeds and in an appropriate manner while keeping a proper lookout to minimize the likelihood of groundings and collisions with fauna.

## 4. Conclusions

This review highlighted several aspects associated with boating and shipping operations, their associated activities and supporting infrastructure that present potential environmental impacts. The acknowledged environmental impacts, which although primarily impact aquatic biomes also includes impacts to atmospheric and terrestrial zones, are varied and include physical, chemical, and biotic factors. This includes elements such as physical changes to bottom substrate and habitats from sources such as anchoring and mooring and vessel groundings, alterations to the physico-chemical properties of the water column and aquatic biota through the application of antifouling paints, operational and accidental discharges (ballast and bilge water, hydrocarbons, garbage, and sewage), and fauna collisions and other disturbances. Additionally, physical, chemical, and biological impacts may occur either directly or indirectly, thus overlapping or transforming between physical, chemical, and biological impact categories. Furthermore, initial and continuing impacts vary on ranging spatial and temporal scales, according to factors such as the size and density (e.g., cumulative impacts of clustered vessels), movement and specific activity-based operations of boats and vessels, and environmental settings and conditions. To ensure that our environmental resources are used sustainably, the impacts associated with boating and shipping operations, their associated activities and supporting infrastructure need to be appropriately managed.

Addressing environmental impacts associated with boating and shipping is a typically varied and challenging task, in addition to impacts operating over varying spatial and temporal scales, and varying user groups with competing needs and believed responsibilities, ambitions, economic interests,

and the potential exploitation of legal disparities and inefficient coordination within and between different (nations) maritime zones. However, various measures do exist to ensure sustainable management of boating- and shipping-related environmental impacts. Broadly these are categorized as direct regulatory, informational and motivational, or economic instruments. Direct regulatory instruments have historically proven relatively effective in reducing environmental destruction and pollution, particularly in instances where there is evidence of gross negligence and willful or careless actions. Unfortunately, they also tend to be relatively inflexible and, if not appropriately designed and implemented, prohibitively costly to administer. Not surprisingly, these instruments have attracted criticism, quite often from target industry. Management instruments designed to complement, or in some cases even substitute, the traditional regulatory approach have been adopted. These supplementary measures offer various levels of industry participation and self-regulation.

No simple solutions exist regarding environmental problems, each type of management measure or approach possesses its own individual advantages and disadvantages toward successful implementation. Furthermore, specific environmental management issues are characteristically unique and complex requiring the adoption and implementation of a range of measures to successfully fulfil their desired management objective. To be effective, environmental managers must seek to develop a comprehensive understanding of the full range of instruments available, and the respective roles they play in helping to achieve positive environmental outcomes, including the pros and cons of the various regulatory alternatives. Following the acquisition of relevant knowledge and understanding, managers may approach specific environmental problems in a clear, logical, and strategic way (especially being able to clearly articulate the problem), while remaining agile, innovative, and open to change and new ideas.

**Author Contributions:** Conceptualization, T.A.B. and R.J.K.D.; data curation, T.A.B.; investigation, T.A.B. and R.J.K.D.; methodology, T.A.B.; supervision, T.A.B.; writing—original draft, T.A.B. and R.J.K.D.; writing—review & editing, T.A.B. and R.J.K.D. All authors have read and agreed to the published version of the manuscript.

**Funding:** This research received no external funding.

**Acknowledgments:** The authors would like to thank the three anonymous reviewers for supportive remarks and constructive comments.

**Conflicts of Interest:** The authors declare no conflict of interest.

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
