# Peer review of "Boating- and Shipping-Related Environmental Impacts and Example Management Measures: A Review"

_jmse, doi:10.3390/jmse8110908_

Round 1

Reviewer 1 Report

The review is really interesting and well written. It covers all the aspects related to the environmental impact of boating and shipping operations. Its also provides a comprehensive overview of a broad range of government and industry related instruments and measures for boating and shipping related environmental impact management. The illustrated management options could be applicable in similar worldwide settings and be  usable by resource managers to better target pollution regulations and programs for improved sustainability of aquatic biomes.

Reviewer 2 Report

This article from Byrnes and Dunn reviews the "Environmental Impacts of Boating and Shipping in coastal water ecosystems". It deals with an important subject and issue in coastal water pollution, which the authors presented as a reviewed study. The acquaintance and understanding of the main impacts of this issue are crucial for their short and long-term monitoring and mitigating. The study evidences the physical and chemical impacts on the water column quality and the aquatic biota derived from the Boating and Shipping operations. The core of the paper provided an exhaustive inventory of the Boating and Shipping Related Environmental Impact. Besides, they present various policy instruments and positive measures implemented by the Australian government and industries to mitigate and better manage the impacts they described, which may be implemented worldwide. Besides, an ensemble of measures is, as well, presented, which is characterized as ‘Common Sense’ Management Measures, but, in reality, are valuable ideas for creating tools for management, mitigation, and reduction of the potential impacts. They are the following: 'behavioral' and 'practical' measures, which individuals, institutions, and governments may adopt for the proposed objectives. The paper is well written and easy to read. In my opinion, it is a valuable and interesting paper for publishing in JMSE, as the theme addresses an important and contemporary environmental issue.

Reviewer 3 Report

At a time of large anthropogenic pressure from boating and shipping operation to marine environment the topic of this paper is very important to broader audience and fall within the Aims and Scope of the journal.

This paper gives a quite well designed overview of all potential boating and shipping environmental impacts with proposed measures to minimize it. 

Although pretty large, the manuscript is well-structured, the title and section headings reflect the content.

Based on extensive literature the paper explain current knowledge about the topics and propose future measures.

The reference list is quite impressive. The recent literature is used and entries in reference list correspond to references in text and vice versa.

Conclusion is well-grounded and logically drawn.

Although, paper is written according the instruction for author some minor technical improvement is needed:

Line 538. At the end of sentences a full stop is needed.

Line 768. At the end of sentences a full stop is needed.
